# Return-Critic: Bridging Goal Discrepancy for Efficient Visual Reinforcement Learning

**Ruyi Lu**[1]  **Xuesong Wang**[1]  **Hengrui Zhang**[1]  **Yuhu Cheng**[1]

## Abstract

Sample inefficiency remains a challenge in pixel-based visual reinforcement learning (RL), primarily due to ineffective state representation learning. While recent advances employ auxiliary tasks to improve representation learning, their representation goals (e.g., mask reconstruction, state prediction) are misaligned with the ultimate RL goal of maximizing return, constraining further improvements in representation quality. To achieve efficient visual reinforcement learning, we propose **Return-Critic (RC)**, an auxiliary framework that bridges goal discrepancy by return prediction. RC samples partial frames from an episode, processes them through a shared visual encoder, and employs a lightweight Transformer to predict the episode's return, forcing the encoder to learn return-relevant representation. The attention weights naturally highlight important frames, enabling a key function for prioritized learning. Extensive experiments on both online (DMControl) and offline (V-D4RL) benchmarks demonstrate that RC significantly enhances the sample efficiency, particularly achieving 68% performance boost on average across nine challenging tasks from DMControl.

## 1. Introduction

Pixel-based visual reinforcement learning (RL) aims to enable agents to learn complex decision-making policies directly from high-dimensional visual observations. This ability has broad applications in real-world scenarios such as robotic manipulation (Panaganti et al., 2024; Hafner et al., 2025) and autonomous driving (Zhang et al., 2025a). However, compared to RL with low-dimensional state vectors,

[1]School of Information and Control Engineering, China University of Mining and Technology, Xuzhou, China. Correspondence to: Yuhu Cheng <chengyuhu@163.com>.

*Proceedings of the 43rd International Conference on Machine Learning*, Seoul, South Korea. PMLR 306, 2026. Copyright 2026 by the author(s).

visual RL generally requires a large number of environment interactions to achieve satisfactory performance. Its low sample efficiency has become a major bottleneck in practical deployment. Therefore, improving sample efficiency in visual RL with limited interaction samples is of significant practical importance.

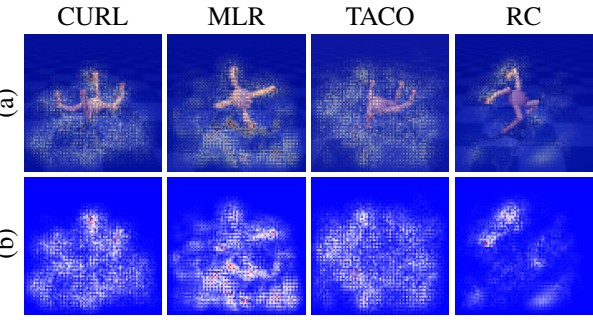

*Figure 1.* Saliency maps of different auxiliary tasks (only running auxiliary tasks). The highlighted part represents the area of task focus. (a) input images with saliency, (b) saliency maps.

The core challenge in visual RL stems from learning decision-relevant state representations from redundant pixel inputs under limited representational capacity. Existing approaches primarily enhance representation learning through two strategies: 1) Pre-trained visual encoders (Yuan et al., 2022; Zhang et al., 2025b), which leverage large-scale image datasets to learn generic visual features; and 2) Self-supervised auxiliary tasks (Zhu et al., 2022; Zheng et al., 2023), which introduce additional goals (e.g., contrastive learning (Laskin et al., 2020b), mask reconstruction (Yu et al., 2022), state prediction (Yu et al., 2021)) jointly optimized with policy learning. Although the information incorporated by these additional goals partially helps agent decision optimization, a goal discrepancy exists between their representation goals and the ultimate RL decision goal of maximizing return. To visually interpret the learned representations, we utilize Guided Backpropagation to project the encoder's last-layer gradients back to the input observation, generating high-resolution saliency heatmaps that highlight the features most causally related to the objective. Figure 1 clearly shows that the previous auxiliary tasks focused on many areas that are not directly related to

the RL goal. This discrepancy causes the representations to capture information that lacks direct relevance to long-horizon return, resulting in only a portion of the introduced information being useful for RL. Crucially, this misalignment allows inefficient information to occupy the limited representational capacity, constraining improvements in representation quality.

Due to limited network capacity, efficient RL representations need capture information directly aligned with the ultimate objective of return maximization. In conventional visual RL (VRL), Q-critics rely on TD learning to predict immediate rewards. While TD updates ideally converge to the true expected return, this assumption frequently fails in VRL. High-dimensional pixel redundancy (de Oliveira et al.; Zhao et al., 2025) and severe error accumulation from TD bootstrapping (Nauman et al., 2024; Wang et al., 2024a) significantly degrade state representation. Empirically, although the n-step TD error ($\sim$1) minimizes quickly during training, a massive gap ($\sim$40) persists in predicting the true long-term cumulative value. This reveals a fundamental issue where TD heavily biases the visual encoder towards local features associated only with proximal rewards. Figure 3 provides intuitive visual evidence for this discrepancy, showing that the TD-driven Q-Critic focuses merely on ground contact points and entirely misses the joints dictating the long-term trajectory. Our Return-Critic mitigates this goal discrepancy by providing a global episodic supervision signal. By explicitly targeting long-range return dependencies overlooked by standard TD updates, Return-Critic injects novel and non-redundant information to maximize the utility of the visual encoder.

In this paper, we propose RC, an auxiliary framework that bridges representation goal and RL goal discrepancy by using episode-level return prediction. RC samples partial frames from an episode, processes them through a shared visual encoder, and employs a lightweight Transformer to predict the episode's return. This design forces the visual encoder to prioritize information with the strong correlation to long-horizon return, optimizing the utilization of constrained representational capacity. Unlike existing auxiliary tasks, RC's return-centric goal is directly related to the ultimate goal of RL, bridging goal discrepancy. Critically, by focusing on long-term return rather than long-term rewards, RC provides complementary learning signals to the Q critic. The attention weights enable a key function to assess the importance of samples for return, ensuring effective learning of critical experiences.

We conduct extensive experiments on online tasks from the DeepMind Control Suite and offline datasets from V-D4RL. The results demonstrate that RC is a flexible, plug-and-play module that can be integrated with various off-policy reinforcement learning algorithms, consistently improving

sample efficiency and outperforming prior state-of-the-art (SOTA) methods in both online and offline settings.

Overall, the contributions of this paper are:

- We propose RC, a novel auxiliary framework bridges the goal discrepancy by return prediction, which can be seamlessly integrated into off-policy RL algorithms.

- We design a return critic that uses a visual encoder and a Transformer to predict the episode's return, guiding the state representations to contain long-term return information.

- We introduce a key function derived from attention weights to assess the importance of state–action pairs and optimize the sampling mechanism.

- Extensive experiments on DMControl and V-D4RL benchmarks show that RC surpasses previous SOTA methods.

## 2. Related Work

### 2.1. Efficient Visual Reinforcement Learning

Learning from pixel observations is a fundamental but challenging problem in visual RL. As a result, efficient visual reinforcement learning has attracted widespread attention, and numerous methods have been designed to leverage limited experience to improve sample efficiency. Model-based methods (Hansen et al., 2022; Wang et al., 2024b; Hafner et al., 2025) explicitly build a world model of the environment to enable planning or facilitate policy learning. Given the nature of pixel input, (Laskin et al., 2020a; Yarats et al., 2021; Hansen & Wang, 2021) utilize data augmentation techniques to enrich the diversity of observations and increase training samples. To obtain richer state representations, approaches (Yuan et al., 2022; Zhang et al., 2025b) pre-train general visual encoders on large-scale real or augmented image datasets. Another mainstream method introduces auxiliary tasks to improve representation learning quality. (Laskin et al., 2020b) pioneered contrastive learning tasks to learn discriminative representations. (Yu et al., 2022; Zhu et al., 2022; Lee et al., 2024) perform latent space reconstruction from masked pixels to learn the spatiotemporal relationships between partial samples. (Schwarzer et al., 2021; Yu et al., 2021; Zheng et al., 2023) capture short-term dynamic information by learning forward or backward state feature predictions. Notably, some studies attempted to incorporate return-related information into representation learning, albeit often in an indirect or localized manner. Indirectly, (Liu et al., 2021a) uses the future return of states as labels for contrastive learning samples. Locally, (Zheng et al., 2023) also predicts the return within a short

horizon. Furthermore, recent advancements have explicitly utilized return signals to condition sequence modeling (Wang et al., 2025) or decouple reward-related features for offline reinforcement learning (Yang et al., 2025). However, these methods often rely on static offline datasets or localized horizons, leaving the fundamental goal discrepancy in continuous online TD learning unaddressed. Unlike above methods, this work aims to design an auxiliary task that directly relates the representation goal to complete return, helping the agent capture long-term return information from observations.

## 2.2. Transformer

The Transformer, with its powerful ability to model long sequences and capture global dependencies, has revolutionized the fields of natural language processing (Ye et al., 2025) and computer vision (Zhu et al., 2025). However, when processing long sequences, Transformers face a computational bottleneck due to the quadratic growth in computational complexity of the self-attention mechanism with sequence length. To address this, researchers have proposed various efficient attention variants (Beltagy et al., 2020; Zaheer et al., 2020; Liu et al., 2021b). Among them, the Longformer architecture introduces a hybrid attention mechanism combining local window attention and global attention, achieving linear complexity. In the visual reinforcement learning field, Transformer has been widely introduced. Some works (Yu et al., 2022; Lee et al., 2024) use it as a tool for representation learning, utilizing Transformer for mask reconstruction or feature prediction; others treat it as a sequence decision model (Chen et al., 2021; Janner et al., 2021), treating reinforcement learning as a sequence modeling problem and directly outputting actions. This work requires modeling long sequences, and considering the timeliness of visual RL, Longformer is selected. Furthermore, this work trains a key function based on attention weights scores from Transformer.

## 3. Methodology

### 3.1. Preliminaries

**Visual Reinforcement Learning.** Reinforcement learning learns policies from state inputs through dynamic interaction with the environment, modeled as a Markov Decision Process (MDP) (Haarnoja et al., 2018) $M = \langle \mathcal{S}, \mathcal{A}, \mathcal{P}, \mathcal{R}, \gamma \rangle$ Here, $\mathcal{S}$ denotes the state space, $s_t \in \mathcal{S}$ is the state obtained from the environment at time $t$, $\mathcal{A}$ denotes the action space, $a_t \in \mathcal{A}$ is the action taken by the agent at time $t$, $\mathcal{P}$ is the state transition function, $\mathcal{R}$ is the reward function, $r_t = \mathcal{R}(s_t, a_t)$ is the reward received for taking action $a_t$ in state $s_t$, and $\gamma \in [0, 1]$ is the discount factor. The MDP also includes an initial state distribution $p_0(s_0)$. The goal of RL is to find an optimal policy $\pi(a_t|s_t)$ that maximizes the

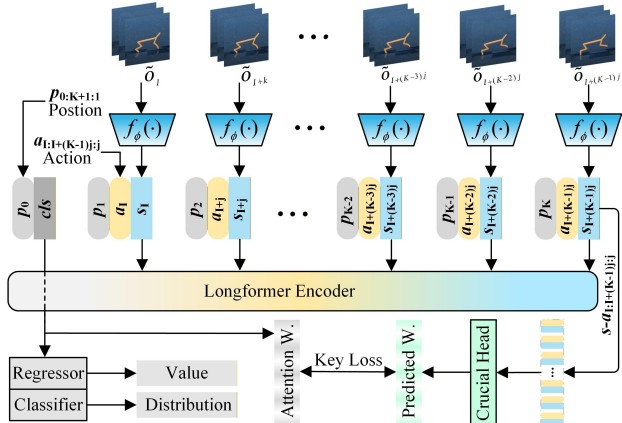

*Figure 2.* RC Framework. Notably, the attention weight scores used to train the key function are derived from the self-attention output of the *cls* token in the first layer of Longformer.

discounted expected return:

$$R = \mathbb{E}_\pi \left[ \sum_{t=0}^{\infty} \gamma^t \mathcal{R}(s_t, a_t) \right] \tag{1}$$

where the expectation is computed over trajectories induced by policy $\pi$ and environment dynamics $\mathcal{P}$.

In visual reinforcement learning scenarios, the environmental state faced by the agent is typically presented as high-dimensional pixel data. Although learning from visual observations is often formulated as a Partially Observable MDP (POMDP) problem, this paper follows the approach in (Liu et al., 2023) by adopting an approximate treatment: considering a sequence of three consecutive image observations as a fully observable state. Let $O$ be the high-dimensional observation space; then the observation state at time $t$ is defined as $\tilde{o}_t = \{o_t, o_{t-1}, o_{t-2}\}$, where $o \in O$. To deeply understand the environment and make optimal decisions, the agent needs to transforms the high-dimensional state $\tilde{o}_t$ into a low-dimensional state representation $s_t$.

### 3.2. Return-Critic

RC bridges the discrepancy between the representation goal and the RL goal by introducing an episode-level return prediction task, enhance representation quality. The gradient direction generated by this return prediction aligns with the information-theoretic objective of capturing return-relevant information, thereby mitigating representation limitations caused by goal discrepancy. In addition, RC predicts return based on partial state-actions within episode, avoiding information overlap with the single state-action reward prediction in policy learning. We illustrate the overall framework of RC in Figure 2 and provide details below.

**Framework.** As shown in Figure 2, RC first sequentially encodes the observation state samples uniformly sample from episode to state representations using a visual encoder.

The state representations are then paired with corresponding actions to form state-action pair tokens. A special *cls* token is prepended to the state-action pair token sequence to aggregate global episode information, and position encoding is applied to the *cls* token and state-action pair tokens. Then, Longformer is used to model dependencies between return and state-action pair tokens, outputting a *cls* token. Finally, this *cls* token is passed through a multi-layer perceptron (MLP) to predict (regress and classify) return of the episode. Additionally, the attention weight scores from the *cls* token are used to train a key function, which predicts the importance of state-action pairs for the return, optimizing the sampling mechanism. Below, we detail the processes of *return prediction*, and *importance prediction*.

**(i) Return Prediction.** A shared CNN-based visual encoder $f_\phi(\cdot)$ sequentially encodes the observation sample sequence $\tau_K^{\tilde{o}} = \{\tilde{o}_I, \tilde{o}_{I+j}, \cdots, \tilde{o}_{I+(K-1)j}\}$ to obtain the state representation sequence $\tau_K^s = \{s_I, s_{I+J}, \cdots, s_{I+(K-1)j}\}$:

$$\tau_K^s = f_\phi(\tilde{o}_I), f_\phi(\tilde{o}_{I+j}), \cdots, f_\phi(\tilde{o}_{I+(K-1)j}) \quad (2)$$

The sequence of observation state samples $\tau_{1+J}^{\tilde{o}}$ is obtained by uniformly sampling from continuous observation frames of an episode. Uniform sampling means selecting a starting point $I$ from a sequence of $n$-frame length, and then taking $K$ samples with a fixed interval $j$. Considering MDP approximation, RC selects the $(I + kj)$-th frame $(k = 0, 1, \cdots, K - 1)$ and its two preceding frames as the observation state sample $\tilde{o}_{I+kj}$. The interval $j$ and sample count $K - 1$ are hyperparameters, and the start point $I$ is a random number within: $2 < I < n - (K-1)j$.

RC uses an efficient Transformer variant (Longformer) to process the entire episode at once. Since the episode return in RL is determined by both the states and actions, RC pairs state representations with corresponding actions as state-action pair tokens. Additionally, RC prepends a special *cls* token to the state-action pair tokens, and Longformer maintains global attention on this *cls* token. As shown in Figure 2, the input to Longformer includes the *cls* token and the state-action pair sequence $\tau_K^{s-a} = \{(s_I, a_I), (s_{I+j}, a_{I+j}), \cdots, (a_{I+(K-1)j}, s_{I+(K-1)j})\}$. The state-action pair sequence is formed from the state representation sequence $\tau_K^s$ and the action sequence $\tau_K^a = \{a_I, a_{I+j}, \cdots, a_{I+(K-1)j}\}$. Then, RC encodes relative temporal position information into the *cls* token and state-action pair tokens:

$$\tau_{K+1}^p = [cls, \tau_K^{s-a}] + [p_0, p_1, \cdots, p_K] \quad (3)$$

The token sequence is passed through a Longformer encoder $g_\psi(\cdot)$. Longformer includes sparse self-attention, layer normalization, and a multi-layer perceptron block with residual connections. After passing through Longformer, the output sequence $x$ is obtained: $x_{K+1} = g_\psi(\tau_{K+1}^p)$.

Finally, RC extracts the *cls* token as output, denoted $c = x_{K+1}[0]$, where $x_{K+1}[0]$ is the first token (*cls* token) in the

output sequence. $c$ will be used for subsequent return prediction heads. The *cls* token $c$ aggregates the spatiotemporal information of the entire episode sequence. Return prediction is achieved through tow prediction heads (regressor and classifier), aligning representation goal with the final goal of RL (maximizing return). Specifically, $c$ is input into a regressor and a classifier separately to predict the value and the discrete distribution of the return.

Regressor goal is to learn a mapping function that maps the *cls* token $c$ to the predicted return value $\hat{R}$. By minimizing $\mathcal{L}_{reg}$, RC learns to extract continuous features related to long-term return from observations. For a batch of size $N$, the regression loss uses the mean squared error (MSE):

$$\mathcal{L}_{reg} = \frac{1}{N} \sum_{q=1}^{N} (R_q - \hat{R}_q)^2 \quad (4)$$

The classifier learns a mapping function that maps the *cls* token $c$ to a class logit vector $\hat{\chi}$. The value range of returns is divided into $M$ intervals, and the label $y$ indicates the interval to which the return belongs. By minimizing $\mathcal{L}_{cls}$, RC can learn return-discriminative features from observations. The classification loss uses the cross-entropy loss:

$$\mathcal{L}_{cls} = -\frac{1}{N} \sum_{q=1}^{N} \sum_{m=1}^{M} y_{q,m} \log(\text{softmax}(\hat{\chi}_q)_m) \quad (5)$$

where $\text{softmax}(\hat{\chi}_q)_m = \frac{\exp(\hat{\chi}_{q,m})}{\sum_{\iota-1}^{M} \hat{\chi}_{q,\iota}}$ converts logits to a probability distribution.

The overall loss function for return prediction is the weighted sum of regression and classification losses:

$$\mathcal{L}_{return} = \lambda_1 \mathcal{L}_{reg} + \lambda_2 \mathcal{L}_{cls} \quad (6)$$

where $\lambda_1$ and $\lambda_2$ are loss weights.

**(ii) Importance Prediction.** The key function is trained using the attention weight scores from the Longformer in the return prediction stage. The design of this importance prediction is inherently grounded in the *Credit Assignment* capability of the Transformer. Specifically, the attention mechanism automatically focuses on discriminative inputs that decisively impact the final return, effectively capturing both critical successful actions (e.g., jumping) and erroneous states (e.g., falling). As visualized in Figure 6, and further validated by recent studies in reasoning models (Liu et al., 2025a), extracting the normalized attention weight score as a pseudo-label provides a theoretically intuitive and empirically effective metric to measure sample importance.

Formally, the key function $\kappa$ aims to map a state-action pair $(s, a)$ to a real number, representing the importance of that pair for the overall return $R$. For a sampled sequence $\tau_K$, the attention weight vector of the *cls* token $c$ towards

all sequence elements is denoted as $\alpha = [\alpha_1, \alpha_2, \ldots, \alpha_K]$. The key function $\kappa_\rho$ is parameterized by an MLP network, taking the state-action pair $(s_k, a_k)$ as input and outputting the predicted key score $\hat{\alpha}_k$. The training objective is to minimize the Mean Squared Error (MSE):

$$\mathcal{L}_{key} = \frac{1}{K} \sum_{k=0}^{K-1} (\alpha_k - \kappa_\rho(s_k, a_k))^2 \qquad (7)$$

To ensure the full method functions reliably and stably in practice, we implement three rigorous mechanisms during the optimization of Eq. (7). First, because the auxiliary return prediction task converges rapidly (empirically within a short span of 10K steps), the target attention distribution $\alpha$ naturally stabilizes very early in training. Second, we strictly apply a *detach* (stop-gradient) operation to the target score $\alpha_k$. This cuts off gradient backpropagation from $\mathcal{L}_{key}$ to the Longformer, preventing the importance prediction objective from disrupting the shared visual encoder. Finally, we employ strict gradient clipping during backpropagation to eliminate risks of numerical overflow or exploding gradients.

To optimize the sampling efficiency, RC introduces Key-Score Prioritized Sampling based on these learned metrics. From the replay buffer, a batch of $B$ samples $\{(s_i, a_i, r_i, s_{i+1})\}_{i=1}^{B}$ is randomly drawn, and their importance is predicted as $\kappa_\rho(s_i, a_i)$. The sampling probability $P(i)$ for each transition is proportional to its key score and is normalized using a softmax function:

$$P(i) = \frac{\exp(\kappa_\rho(s_i, a_i)/\mu)}{\sum_{i=1}^{B} \exp(\kappa_\rho(s_i, a_i)/\mu)} \qquad (8)$$

where $\mu$ is the temperature parameter controlling the prioritization strictness.

### 3.3. Theoretical Analysis

Our analysis is grounded in the information bottleneck view of representation learning, providing a principled understanding of how goal alignment improves representation quality in visual RL. We formalize the notion of return-relevant information and study how goal discrepancy limits the amount of such information that can be captured under limited representational capacity. We further show that RC reduces this discrepancy by explicitly learning to predict episode return, yielding encoder gradients that align with an information-theoretic objective. The saliency maps in Figure 1 and Figure 3 empirically validate our theoretical analysis, demonstrating that well-trained RC focuses on relatively concise and return-relevant regions. All proofs are provided in Appendix A.

**Return-relevant information.** Following the RC classification head in Section 3.2, we discretize the episode return $R$ into a categorical label

$$Y \triangleq b(R) \in \{1, \ldots, M\} \qquad (9)$$

where $b(\cdot)$ partitions the return range into $M$ bins. Let $s_t = f_\phi(\tilde{o}_t) \in \mathbb{R}^d$ be the visual representation. To quantify how much return-relevant information is contained in the learned representation, we consider an episode-level summary

$$z_\phi(\tau) \triangleq \frac{1}{T} \sum_{t=0}^{T-1} \eta(s_t, a_t) \qquad (10)$$

where $\eta(\cdot)$ is a fixed embedding of state-action pairs (e.g., the token embedding used before feeding tokens into the Transformer). We measure representation quality by the mutual information $I(z_\phi; Y) = H(Y) - H(Y \mid z_\phi)$.

**Information objective and goal discrepancy.** Define the return-information objective

$$\mathcal{F}(\phi) \triangleq H(Y \mid z_\phi) \qquad (11)$$

Under limited representational capacity $d$, let $\mathcal{H}_d$ denote the encoder hypothesis class (architecture + embedding dimension $d$), and define the capacity-limited optimal encoder

$$\phi^\star \in \arg \min_{\phi \in \mathcal{H}_d} \mathcal{F}(\phi) \qquad (12)$$

For any auxiliary representation learning objective $\mathcal{L}_{\text{aux}}(\phi)$, we define the (squared) goal discrepancy as

$$\Delta(\phi) \triangleq \mathbb{E}\left[\left\|\nabla_\phi \mathcal{L}_{\text{aux}}(\phi) - \nabla_\phi \mathcal{F}(\phi)\right\|_2^2\right] \qquad (13)$$

Intuitively, $\Delta(\phi)$ quantifies how much the auxiliary learning signal deviates from the ideal direction that directly reduces $H(Y \mid z_\phi)$, i.e., increases return-relevant information.

**Theorem 3.1** (Goal Discrepancy and Representation Quality)**.** *Under limited representational capacity $d$, let $\hat{\phi}$ be an (approximately) stationary solution of the auxiliary goal $\mathcal{L}_{\text{aux}}$ (formalized in Appendix A). Then the return-relevant information captured by the learned representation satisfies*

$$I(z_{\hat{\phi}}; Y) \geq I(z_{\phi^\star}; Y) - \beta\big(\Delta(\hat{\phi}) + \delta^2\big) \qquad (14)$$

*where $\beta > 0$ is a constant determined by the local geometry of $\mathcal{F}$ (PL constant), and $\delta$ quantifies the stationarity accuracy of $\hat{\phi}$.*

Theorem 3.1 shows that goal discrepancy directly limits the *guaranteed* amount of return-relevant information that the learned representation can capture relative to the best encoder in the capacity-limited class $\mathcal{H}_d$. When the auxiliary objective is misaligned with $\mathcal{F}$, the induced representation update may encode task-irrelevant factors, which increases $H(Y \mid z_\phi)$ and reduces $I(z_\phi; Y)$. The proof is given in Appendix A (Theorem .4).

Table 1. Comparison of return scores for different algorithms on 6 common DMControl tasks at 100K and 500K environment steps. "w/ RC" denotes integration with RC. Best results in bold, second-best underlined.

| DMCONTROL (100K STEP) | PLAY VIRTUAL | MLR | TACO | RESACT | DRQ | DRQ W/ RC | DRQV2 | DRQV2 W/ RC |
|---|---|---|---|---|---|---|---|---|
| CARTPOLE SWINGUP | 816±36 | 806±48 | 782±51 | 819±44 | 759±92 | **871±12** | 612±82 | 867±9 |
| REACHER EASY | 785±142 | 866±103 | 821±97 | 917±59 | 601±213 | **923±47** | 481±125 | 883±37 |
| CHEETAH RUN | 474±50 | 482±38 | 402±62 | 503±42 | 344±67 | **564±55** | 372±46 | 531±22 |
| WALKER WALK | 460±173 | 643±114 | 601±103 | 772±65 | 612±164 | **891±43** | 572±43 | 874±24 |
| FINGER SPIN | 915±49 | 907±58 | 876±67 | **974±42** | 901±104 | 964±22 | 625±88 | 943±7 |
| BALL IN CUP CATCH | 929±31 | 933±16 | 902±54 | 948±44 | 913±53 | **960±13** | 687±104 | 947±6 |
| **AVERAGE** | 729.8 | 772.8 | 730.7 | 822.1 | 688.3 | **862.8** | 618.2 | 840.8 |
| 500K STEP | | | | | | | | |
| CARTPOLE SWINGUP | 865±11 | 872±5 | 870±21 | 870±12 | 868±10 | 873±7 | 869±7 | **875±3** |
| REACHER EASY | 942±66 | 957±41 | 944±50 | **974±16** | 942±71 | 965±6 | 924±21 | 973±7 |
| CHEETAH RUN | 719±51 | 674±37 | 663±30 | 750±8 | 660±96 | 804±8 | 672±63 | **817±38** |
| WALKER WALK | 928±30 | 939±10 | 914±87 | 953±21 | 921±45 | 946±12 | 883±94 | **963±15** |
| FINGER SPIN | 963±40 | 973±31 | 972±89 | **979±4** | 938±103 | 976±9 | 897±38 | 966±16 |
| BALL IN CUP CATCH | 967±5 | 964±14 | 960±22 | 967±4 | 963±9 | 973±7 | 919±8 | **977±4** |
| **AVERAGE** | 897.3 | 896.5 | 887.1 | 915.5 | 882.0 | 923.1 | 860.7 | **928.6** |

**RC reduces goal discrepancy via return prediction.** RC learns to predict the return label $Y$ from a uniformly sampled subsequence of $K$ state-action tokens. Let

$$z_{\phi,K}(\tau) \triangleq \frac{1}{K} \sum_{k=1}^{K} \eta(s_{t_k}, a_{t_k}) \qquad (15)$$

where $\{t_k\}_{k=1}^{K}$ are uniformly sampled indices. RC uses Longformer $g_\psi(\cdot)$ with a global *cls* token to aggregate sampled tokens, and a classifier head to output $q_{\omega_c}(Y \mid c)$, trained with cross-entropy loss. Denote the (stochastic) gradient estimator of the RC classification loss with respect to the encoder parameters by $\hat{g}_{\text{RC}}$.

**Theorem 3.2** (Goal Alignment via Return Prediction (RC)). *Define the RC discrepancy to the ideal information gradient:*

$$\Delta_{\text{RC}}(\phi) \triangleq \mathbb{E}\left[\left\|\hat{g}_{\text{RC}} - \nabla_\phi \mathcal{F}(\phi)\right\|_2^2\right] \qquad (16)$$

*Then, under mild regularity assumptions (Appendix A), RC satisfies*

$$\Delta_{\text{RC}}(\phi) \leq \mathcal{O}\left(\epsilon^2 + \frac{1}{K}\right) + V \qquad (17)$$

*where $\epsilon^2$ is the RC predictor's excess prediction/training error, $K$ is the number of sampled observations per episode, $\mathcal{O}$ is asymptotic upper bound, and $V$ upper-bounds the mean-squared stochastic gradient noise of $\hat{g}_{\text{RC}}$.*

Theorem 3.2 explains why RC effectively mitigates goal discrepancy: the discrepancy is controlled by (i) the prediction/training error $\epsilon$ of the return predictor, (ii) the subsampling approximation error $1/K$, and (iii) stochastic gradient noise $V$. With sufficient predictor training ($\epsilon \to 0$) and adequate sampling ($K$ large), RC yields encoder updates that

closely track the ideal return-information gradient $\nabla_\phi \mathcal{F}(\phi)$, thereby improving the guarantee in Theorem 3.1. The full proof is given in Appendix A (Theorem .12).

# 4. Experiment

This section evaluates the sample efficiency of RC in both online RL and offline RL settings. To comprehensively assess RC's sample efficiency in online RL, we tested it on six commonly used and nine more challenging continuous visual control tasks in the DeepMind Control Suite (DMControl) (Tunyasuvunakool et al., 2020). For offline RL, we validated the method's general effectiveness on three tasks in the V-D4RL (Lu et al., 2022) benchmark. We then conducted ablation experiments to further understand the design choices of RC.

## 4.1. Comparison of Return-Critic and Strong Baselines in Online RL Tasks

**Environment Setup:** In the online RL experiments, we first selected six commonly used continuous visual control tasks from the DMControl suite to evaluate the sample efficiency of RC. To further test the algorithm's efficiency in more challenging scenarios, we continued testing on nine more difficult tasks within the same platform. These tasks require the agent to master complex motion control skills and high-precision environmental interaction abilities, often involving delayed or sparse rewards. Thus, existing visual RL algorithms have yet to fully master these tasks, and learning effective strategies requires carefully balancing exploration and exploitation while addressing these challenges.

**Baselines:** For the six common tasks in DMControl, we selected the following algorithms as baselines: (1) DrQ

*Table 2.* Comparison of return scores for DrQv2 w/ RC and other algorithms after 1 million environment steps. The best results are highlighted in bold, and the second-best results are underlined.

| DMControl (1M Step) | DRQ | DREAMERV3 | TACO | RESACT | MIND | DRQV2 | DRQV2 w/ **RC** |
|---|---|---|---|---|---|---|---|
| QUADRUPED RUN | 179±18 | 331±42 | 541±38 | 374±36 | 522±30 | 407±21 | **714±52** |
| HOPPER HOP | 192±41 | **369±21** | 261±52 | 233±32 | 287±9 | 189±35 | 340±43 |
| WALKER RUN | 451±73 | **765±32** | 637±11 | 554±21 | 663±22 | 517±43 | 736±14 |
| QUADRUPED WALK | 120±17 | 353±27 | 793±8 | 690±128 | 687±54 | 680±52 | **875±15** |
| CHEETAH RUN | 474±32 | 728±32 | 821±48 | 792±7 | 746±16 | 691±42 | **897±12** |
| FINGER TURN HARD | 91±9 | 810±58 | 632±75 | 857±80 | 512±49 | 220±21 | **913±55** |
| ACROBOT SWINGUP | 24±8 | 210±12 | 241±21 | 167±25 | 228±30 | 128±8 | **450±70** |
| REACHER HARD | 471±45 | 499±51 | 883±63 | 638±44 | **955±21** | 572±51 | 913±45 |
| REACH DUPLO | 36±7 | 119±30 | **234±21** | 102±26 | 141±24 | 206±32 | 228±7 |
| **AVERAGE** | 226.4 | 464.9 | 560.3 | 489.7 | 526.7 | 401.1 | **673.9** |

(Yarats et al., 2021) introduces data augmentation based on SAC algorithm; (2) DrQv2 (Yarats et al., 2022) enhances data augmentation and optimization details based on DDPG; (3) PlayVirtual (Yu et al., 2021) uses visual trajectory generation for data augmentation; (4) MLR (Yu et al., 2022) reconstructs spatiotemporal masked images; (5) TACO (Zheng et al., 2023) introduces contrastive learning to predict states and return within short horizon; (6) ResAct (Liu et al., 2025b) designs residual actions for incremental motion adjustments. For the nine challenging tasks in DMControl, we include the baselines (1)-(4) as well as (7) Dreamerv3 (Hafner et al., 2025) learns a world model for action selection, and (8) MIND (Lee et al., 2024) cooperates mask reconstruction and action discrimination tasks. All experiments used five random seeds and consisted of 10 evaluation runs per performance test. To directly measure sample efficiency, we report the average return score and standard deviation across these runs. The final algorithm performance is evaluated using the average return score of each task at a fixed step length.

**For the six common tasks in DMControl:** To comprehensively demonstrate the universality of RC across different algorithms, this section applies RC to both the SAC-based DrQ algorithm and the DDPG-based DrQv2 algorithm for benchmarking. We follow the original experimental setups of these algorithms for training and evaluation. In Table 1, we can see that although DrQv2 is an upgraded version of DrQ, DrQv2 has lower return scores than DrQ at the 100K short step length, while at the 500K step length, their return scores are close. This is because DrQ, relying on SAC's entropy maximization mechanism, can converge quickly in the early stages of training. Although DrQv2 enhances long-term learning capability for complex tasks by expanding the experience replay buffer, the larger buffer reduces the utilization of new experience samples, leading to a relative decline in DrQv2's sample efficiency during the initial phase. After introducing RC, both DrQ and DrQv2 (DrQ w/ RC and DrQv2 w/ RC) show significant improvements

in return scores at 100K and 500K step lengths compared to their original versions. Notably, DrQv2 w/ RC's return score at 100K is close to DrQ w/ RC's, and at 500K, it even surpasses DrQ w/ RC. This phenomenon indicates that the key function in RC effectively optimizes experience usage, mitigating the short-term sample inefficiency caused by the large replay buffer. Crucially, DrQ w/ RC and DrQv2 w/ RC outperform other SOTA algorithms at both 100K and 500K step lengths. These results suggest that, compared to other auxiliary tasks, RC further enhances the quality of representations by setting auxiliary tasks aligned with the RL goal, thereby improving sample efficiency.

**For the nine challenging tasks in DMControl:** Given DrQv2's superior convergence stability in long-term training owing to its large-capacity replay buffer, we integrated RC with DrQv2 (denoted as DrQv2 w/ RC) to conduct benchmark tests on nine challenging tasks. As shown in Table 2, DrQv2 w/ RC surpasses the current SOTA in terms of return scores on most tasks and exceeds the suboptimal TACO (predict short-term return) by 20.2% in terms of average return. Compared to the original DrQv2, DrQv2 w/ RC achieves significant improvements across all nine tasks, with an average return increase of 68.0%. This indicates that the success of RC is not merely reliant on DrQv2; rather, it significantly enhances the agent's learning efficiency and final performance in complex visual control tasks by compensating for the lack of long-term return correlations and temporal continuity in existing representations through return critic. Although DrQv2 w/ RC performs excellently in most tasks, it does not reach SOTA levels in a few tasks. Nevertheless, its return scores are suboptimal and close to the best results. This suggests that different tasks may have varying requirements for state representation. By aligning representation goal with RL goal, RC maintains strong adaptability and sample efficiency across diverse task demands.

*Table 3.* The offline evaluation return scores of DrQv2+BC w/ RC and other offline RL methods. Evaluation return scores are mapped from [0, 1000] to [0, 100]. The best results are highlighted in bold, and the second-best results are underlined.

| DMCONTROL | | DATA MEAN | LOMPO | BC | CQL | OFFLINE DV2 | DRQV2+BC | DRQV2+BC w/ RC |
|---|---|---|---|---|---|---|---|---|
| WALKER WALK | MEDIUM | 44.0 | 43.4±11.1 | 40.9±3.1 | 14.8±16.1 | 34.1±19.7 | 46.8±2.3 | **57.4±1.8** |
| | MEDEXP | 70.4 | 39.2±19.5 | 47.7±3.9 | 56.4±38.4 | 43.9±34.4 | 86.4±5.6 | **91.1±4.8** |
| | EXPERT | 97.0 | 5.3±7.7 | **91.5±3.9** | 89.6±6.0 | 4.8±0.6 | 68.4±7.5 | 76.7±6.9 |
| CHEETAH RUN | MEDIUM | 52.4 | 16.4±8.3 | 51.6±1.4 | 40.9±5.1 | 17.2±3.5 | 53.0±3.0 | **59.2±2.6** |
| | MEDEXP | 70.7 | 11.9±1.9 | 57.5±6.3 | 20.9±5.5 | 10.4±3.5 | 50.6±8.2 | **64.0±6.5** |
| | EXPERT | 89.1 | 14.0±3.8 | **67.4±6.8** | 61.5±4.3 | 10.9±3.2 | 34.5±8.3 | 51.5±9.7 |
| HUMANOID WALK | MEDIUM | 57.3 | 0.1±0.0 | 13.5±4.1 | 0.1±0.0 | 0.2±0.1 | 6.2±2.4 | **13.9±2.4** |
| | MEDEXP | 71.6 | 0.2±0.0 | 17.2±4.7 | 0.1±0.0 | 0.1±0.0 | 7.0±2.3 | 14.6±2.3 |
| | EXPERT | 85.8 | 0.1±0.0 | **6.1±3.7** | 1.6±0.5 | 0.2±0.1 | 2.7±0.9 | 5.1±2.4 |

## 4.2. Effectiveness of Return-Critic in Offline RL Tasks

In this section, we discuss the experimental results of RC in offline reinforcement learning settings, highlighting the benefits of Return Critic in visual offline RL. Offline visual RL presents unique challenges as the algorithms must learn the optimal strategy from a fixed dataset without further interaction with the environment. RC can be easily integrated as a plug-and-play module on top of existing visual offline RL methods. For standardized evaluation, we benchmarked on three DMControl visual control tasks from the V-D4RL dataset. In these tasks, the baseline strategy for data collection is a SAC-based policy that captures proprioception state information. For each task, three types of dataset settings were selected: (1) medium; (2) expert; (3) medium-expert (medexp): concatenation of medium and expert datasets. This section compares the return scores of DrQv2+BC w/ RC against other baseline methods. The baseline algorithms and data are all sourced from V-D4RL (Lu et al., 2022).

Table 3 presents the normalized return scores of different algorithms across various datasets, demonstrating that integrating RC with the baseline DrQv2+BC framework consistently enhances overall performance. Specifically, this integration achieves new state-of-the-art results in more than half of the evaluated visual offline RL tasks. Notably, the vanilla base method inherently suffers from severe performance degradation on strictly expert datasets due to the well-documented limitations of simple Behavioral Cloning constraints under limited data coverage. The addition of our RC framework partially alleviates this inherent vulnerability and yields consistent empirical improvements even in these highly challenging scenarios. Furthermore, the results show that RC generally achieves higher scores on mixed medium-expert datasets compared to pure expert datasets. This phenomenon indicates that RC explicitly benefits from diverse data quality, since varying trajectories naturally provide the essential return variance required to perform comprehensive and accurate return predictions. Consequently, this variance enables the visual encoder to learn state representations with

*Table 4.* Return scores after individually removing each core loss function of RC. The best results are highlighted in bold.

| DMCONTROL (1M STEP) | DRQV2 w/ RC | w/o $\mathcal{L}_{reg}$ | w/o $\mathcal{L}_{cls}$ | w/o $\mathcal{L}_{key}$ |
|---|---|---|---|---|
| QUADRUPED RUN | 714±20 | **716±21** | 645±47 | 614±46 |
| CHEETAH RUN | 897±12 | 842±31 | **904±7** | 869±32 |
| WALKER RUN | **736±52** | 668±58 | 651±13 | 654±28 |
| **AVERAGE** | **782.3** | 742.0 | 734.7 | 712.3 |

robust long-term return discriminability. Ultimately, while RC does not completely eliminate the offline coverage flaws of the base algorithm, it strictly improves the representation quality and final task scores across all evaluated datasets, thereby firmly underscoring its general applicability as a highly effective plug-and-play module for both online and offline learning frameworks.

## 4.3. Ablation

**Impact of RC loss functions.** This section performs ablation experiments to analyze the contributions of the three core loss functions in RC, including: (1) the regression loss; (2) the classification loss; and (3) the key evaluation loss.

Each loss term's impact is presented in Table 4, showing how a single loss function impacts the return scores. Removing the regression loss decreases scores in tasks like Cheetah Run and Walker Run, while Quadruped Run slightly improves. This is because Walker Run relies more on balance control, where the classification loss helps in identifying impending imbalance states. Removing the classification loss causes a decline in Walker Run and Quadruped Run but an increase in Cheetah Run. This suggests that the classification loss, which emphasizes discriminative features, somewhat limits the agent's ability to learn fine details from observations, thereby affecting continuous reward feature capture. Importantly, removing the key loss results in a performance drop across all tasks, with Cheetah Run showing the smallest decrease, as the task has already nearly con-

*Table 5.* Return scores of RC and DrQv2 equipped with different sampling strategies at 1M steps. KSS consistently outperforms traditional TD error-based sampling methods across all evaluated tasks.

| ENVIRONMENT | RC w/ | | | | DRQV2 w/ | | | |
|---|---|---|---|---|---|---|---|---|
| | KSS | PER | LAP | PAL | KSS | PER | LAP | PAL |
| WALKER RUN | **714±20** | 646±20 | 687±15 | 677±16 | **580±20** | 509±22 | 537±15 | 542±27 |
| CHEETAH RUN | **897±12** | 822±14 | 843±11 | 845±14 | **764±14** | 678±28 | 701±17 | 696±21 |
| QUADRUPED RUN | **736±52** | 658±21 | 679±18 | 692±23 | **457±19** | 423±15 | 443±21 | 437±11 |
| AVERAGE | **782.3** | 708.7 | 736.3 | 738.0 | **600.3** | 536.6 | 560.3 | 558.3 |

verged by the 1M step and the optimized sampling mechanism no longer provides a significant advantage.

**Universality and Efficacy of Key Score Sampling.** To isolate the independent contribution of Key Score Sampling (KSS), we conducted comprehensive cross-ablation experiments across various sampling strategies, including standard Uniform, PER (Schaul et al., 2015), LAP, and PAL (Fujimoto et al., 2020). As summarized in Table 5, integrating KSS yields a substantial performance boost of approximately ten percent for both RC and the baseline DrQv2, significantly outperforming the marginal or negligible gains of traditional methods. This superiority stems from fundamental mechanistic differences. Whereas conventional approaches rely on single-step TD errors and risk overfitting to uninformative high-error transitions, KSS leverages the inherent credit assignment capability of the Transformer to prioritize state-action pairs based on their direct criticality to the overall episodic return.By accurately capturing both decisive actions and critical mistakes, KSS significantly enhances sample efficiency. To prevent the value network from biasing towards outliers, KSS is applied to exactly half of the sampled batch alongside standard uniform sampling. Furthermore, the KSS mechanism functions as a universal module applicable to diverse off-policy algorithms regardless of the input modality. Notably, even when applying a stop-gradient operation to state representations, relying solely on KSS to prioritize the replay buffer achieves significant improvements over the baseline, thereby robustly validating its universality across both high-dimensional visual domains and fully observable low-dimensional environments.

**Saliency maps of return critic and Q critic.** To determine if Return Critic provides unique information to state representations, we employed Guided Backpropagation (Springenberg et al., 2015), a technique that visualizes network attention by highlighting input regions most influential for specific outputs. We generated saliency maps for both Return Critic (return prediction) and Q Critic (reward prediction) across Cheetah Run and Quadruped Run.

Figure 3 reveals distinct attention patterns between the two critics. As shown in the two tasks, Q Critic predominantly

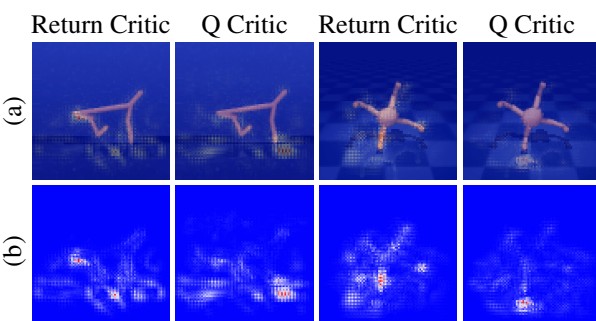

*Figure 3.* Saliency maps of return critic and Q critic. (a) input images with saliency, (b) saliency maps.

focuses on ground contact regions, as predicting immediate rewards only requires evaluating the current state's reward-relevant features. In contrast, Return Critic concentrates on the robot's articulation joints, as predicting cumulative return necessitates understanding how current states influence long-term trajectories. These results demonstrate that Return Critic captures complementary information beyond what Q Critic provides, focusing on features that influence return rather than immediate rewards.

## 5. Conclusions

This paper introduces the Return-Critic (RC), an auxiliary task framework that improves representation learning in visual reinforcement learning. RC bridges the discrepancy between the representation goal and the RL goal by introducing an episode-level return prediction task. Theoretically, RC can be shown to bridge goal discrepancy, thereby improving representation quality. Moreover, RC compensates for the lack of long-term return correlations and temporal continuity in the representations of existing methods. It also trains a key function based on attention weights to optimize the sampling mechanism. Experiments show that RC integrates well with off-policy algorithms, achieving improvements in both online and offline RL tasks. While RC improves sample efficiency, its computational cost depends on sequence length. Future work will focus on improving attention mechanisms for better scalability.

## Acknowledgements

The authors wish to acknowledge and thank the financial support of the National Natural Science Foundation of China under Grants 62573416 and 62373364. We would also like to thank the anonymous reviewers and Area Chairs for their constructive feedback and helpful discussions, which significantly improved the quality of this paper.

## Impact Statement

This paper presents work whose goal is to advance the field of Machine Learning. There are many potential societal consequences of our work, none which we feel must be specifically highlighted here.

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

# Appendix A    Proofs (Notation Aligned with Methodology)

## A.1    Symbols, Definitions, and Alignment to the Main Text

**Given symbols from the main text.**    We use the same modules as in the Methodology:

- **Visual encoder:** $f_\phi(\cdot)$ (CNN-based), with parameters $\phi$.

- **Longformer encoder:** $g_\psi(\cdot)$, with parameters $\psi$.

- **Key function:** $\kappa_\rho(\cdot)$, with parameters $\rho$ (not used in the proofs below).

**Heads (defined here for completeness).**    Given the Longformer *cls* embedding $c$, we define:

- **Regression head:** $\hat{R} = h_{\omega_r}^{\mathrm{reg}}(c)$ (predicts a scalar return).

- **Classification head:** logits $\hat{\chi} = h_{\omega_c}^{\mathrm{cls}}(c) \in \mathbb{R}^M$, and

$$q_{\omega_c}(y \mid c) \triangleq \mathrm{softmax}(\hat{\chi})_y, \quad y \in \{1, \dots, M\}.$$

We denote all RC predictor parameters by $\omega \triangleq (\psi, \omega_r, \omega_c)$, and for Theorem .12 we only need $(\psi, \omega_c)$.

**Episode, return, and return label.**    An episode is $\tau = \{(\tilde{o}_t, a_t, r_t)\}_{t=0}^{T-1}$ where $\tilde{o}_t = (o_t, o_{t-1}, o_{t-2})$. The discounted episode return is $R(\tau) = \sum_{t=0}^{T-1} \gamma^t r_t$. Following the RC classifier design in the main text, define a discretized return label

$$Y \triangleq b(R(\tau)) \in \{1, 2, \dots, M\}, \tag{18}$$

where $b(\cdot)$ bins returns into $M$ intervals/classes. Since $Y$ is discrete, all entropies and mutual informations below are discrete.

**State representations and sampled state-action tokens (same as Methodology).**    The encoder produces state representations

$$s_t = f_\phi(\tilde{o}_t) \in \mathbb{R}^d. \tag{19}$$

RC uniformly samples $K$ indices $\mathcal{I}_K = \{t_1, \dots, t_K\} \subset \{0, \dots, T-1\}$ from the episode and forms state-action tokens $\{(s_{t_k}, a_{t_k})\}_{k=1}^K$. As in the main text, these tokens are prepended with a *cls* token and positional encodings to form $\tau_{K+1}^p$, which is then fed into the Longformer:

$$x_{K+1} = g_\psi(\tau_{K+1}^p), \qquad c \triangleq x_{K+1}[0]. \tag{20}$$

**A tractable episode summary used for the sampling bound.**    To obtain an explicit $\mathcal{O}(1/K)$ term, we analyze a fixed pooling summary of sampled state-action tokens. Let $\eta(\cdot)$ be a fixed measurable embedding of a state-action pair (e.g., the token embedding before attention). Define the sampled summary

$$z_{\phi,K}(\tau) \triangleq \frac{1}{K} \sum_{k=1}^K \eta(s_{t_k}, a_{t_k}) \in \mathbb{R}^m, \tag{21}$$

and the full-episode summary

$$z_\phi(\tau) \triangleq \frac{1}{T} \sum_{t=0}^{T-1} \eta(s_t, a_t). \tag{22}$$

**Remark.** The implemented *cls* token of $g_\psi$ is a learnable global aggregator of sampled tokens. Mean pooling is a special case that can be represented by a sufficiently expressive aggregator, hence analyzing $z_{\phi,K}$ yields a conservative but tractable bound.

**Return-relevant information objective.** We measure representation quality by the conditional entropy of the return label:

$$\mathcal{F}(\phi) \triangleq H(Y \mid z_\phi), \qquad \mathcal{F}_K(\phi) \triangleq H(Y \mid z_{\phi,K}). \tag{23}$$

The mutual information identity gives

$$I(z_\phi; Y) = H(Y) - H(Y \mid z_\phi) = H(Y) - \mathcal{F}(\phi). \tag{24}$$

**Capacity-limited optimum (represents "limited representational capacity $d$").** Let $\mathcal{H}_d$ be the hypothesis class of encoders $f_\phi : \tilde{\mathcal{O}} \to \mathbb{R}^d$ (architecture + embedding dimension). Define the best encoder under this capacity:

$$\phi^\star \in \arg\min_{\phi \in \mathcal{H}_d} \mathcal{F}(\phi). \tag{25}$$

**Goal discrepancy (stable, rigorous definition).** For any auxiliary representation objective $\mathcal{L}_{\mathrm{aux}}(\phi)$, define

$$\Delta(\phi) \triangleq \mathbb{E}\Big[\big\|\nabla_\phi \mathcal{L}_{\mathrm{aux}}(\phi) - \nabla_\phi \mathcal{F}(\phi)\big\|_2^2\Big]. \tag{26}$$

**New symbols introduced in this appendix (meaning).**

- $T$: episode length (finite horizon used in the analysis).

- $K$: number of uniformly sampled state-action tokens per episode in RC.

- $M$: number of return bins/classes for the RC classifier (same meaning as in the main text classifier).

- $d$: representation dimension (output dimension of $f_\phi$).

- $m$: dimension of the token embedding $\eta(s, a)$.

- $\eta(\cdot)$: fixed embedding map used to define $z_\phi$ and $z_{\phi,K}$.

- $\mathbb{D}$: fixed episode distribution (e.g., replay buffer distribution).

- $\mu_F$: PL constant for $\mathcal{F}$ in Assumption .2.

- $\delta$: auxiliary stationarity tolerance in Assumption .3.

- $\epsilon^2$: RC predictor excess cross-entropy (optimization/statistical) error in Assumption .8.

- $B$: uniform bound on token embeddings $\|\eta(s, a)\|_2 \leq B$ in Lemma .7.

- $L_{\phi\omega}$: Lipschitz constant controlling how $\nabla_\phi \mathcal{L}_{\mathrm{cls}}$ changes w.r.t. predictor parameters $\omega$.

- $\mu_\omega$: quadratic-growth/error-bound constant relating predictor excess risk to parameter distance.

- $L_{\mathrm{sam}}$: stability constant linking $\|\nabla_\phi \mathcal{F}_K - \nabla_\phi \mathcal{F}\|$ to $\|z_{\phi,K} - z_\phi\|$.

- $V$: bound on the mean-squared error of the stochastic gradient estimator (to avoid conflict with class count $M$).

## A.2 Assumptions

**Assumption .1** (Fixed episode distribution (off-policy view)). Episodes $\tau$ are sampled from a fixed distribution $\mathbb{D}$ that does not depend on $\phi$.

**Assumption .2** (Local PL condition for $\mathcal{F}$). There exist $\mu_F > 0$ and a neighborhood $\mathcal{N}$ of $\phi^\star$ such that for all $\phi \in \mathcal{N}$,

$$\mathcal{F}(\phi) - \mathcal{F}(\phi^\star) \leq \frac{1}{2\mu_F} \big\|\nabla_\phi \mathcal{F}(\phi)\big\|_2^2. \tag{27}$$

**Assumption .3** (Approximate stationarity of auxiliary optimization). Let $\hat{\phi}$ be obtained by optimizing $\mathcal{L}_{\mathrm{aux}}$. Assume $\hat{\phi} \in \mathcal{N}$ and

$$\mathbb{E}\big[\|\nabla_\phi \mathcal{L}_{\mathrm{aux}}(\hat{\phi})\|_2^2\big] \leq \delta^2. \tag{28}$$

### A.3    Theorem 1 (Aligned): Goal Discrepancy Limits Representation Quality

**Theorem .4** (Goal discrepancy and representation quality (capacity-limited, rigorous)). *Under Assumptions .1–.3, with $\phi^\star$ defined in* (25)*, we have*

$$I(z_{\hat{\phi}};Y) \geq I(z_{\phi^\star};Y) - \frac{1}{\mu_F}\Big(\Delta(\hat{\phi}) + \delta^2\Big). \tag{29}$$

*Proof.* By (24),

$$I(z_\phi;Y) = H(Y) - \mathcal{F}(\phi),$$

hence

$$I(z_{\hat{\phi}};Y) - I(z_{\phi^\star};Y) = -\big(\mathcal{F}(\hat{\phi}) - \mathcal{F}(\phi^\star)\big). \tag{30}$$

By the PL condition (27),

$$\mathcal{F}(\hat{\phi}) - \mathcal{F}(\phi^\star) \leq \frac{1}{2\mu_F}\|\nabla_\phi\mathcal{F}(\hat{\phi})\|_2^2. \tag{31}$$

Using the decomposition

$$\nabla_\phi\mathcal{F}(\hat{\phi}) = \big(\nabla_\phi\mathcal{F}(\hat{\phi}) - \nabla_\phi\mathcal{L}_{\mathrm{aux}}(\hat{\phi})\big) + \nabla_\phi\mathcal{L}_{\mathrm{aux}}(\hat{\phi}),$$

and $(a+b)^2 \leq 2a^2 + 2b^2$, we obtain

$$\|\nabla_\phi\mathcal{F}(\hat{\phi})\|_2^2 \leq 2\|\nabla_\phi\mathcal{F}(\hat{\phi}) - \nabla_\phi\mathcal{L}_{\mathrm{aux}}(\hat{\phi})\|_2^2 + 2\|\nabla_\phi\mathcal{L}_{\mathrm{aux}}(\hat{\phi})\|_2^2.$$

Taking expectation and using (26) and Assumption .3 yields

$$\mathbb{E}\|\nabla_\phi\mathcal{F}(\hat{\phi})\|_2^2 \leq 2\Delta(\hat{\phi}) + 2\delta^2.$$

Substitute into (31):

$$\mathcal{F}(\hat{\phi}) - \mathcal{F}(\phi^\star) \leq \frac{1}{2\mu_F}(2\Delta(\hat{\phi}) + 2\delta^2) = \frac{1}{\mu_F}\big(\Delta(\hat{\phi}) + \delta^2\big).$$

Finally, combine with (30) to obtain (29). $\qquad\square$

### A.4    Model-Aligned Realizability and Sampling Lemmas

**Lemma .5** (Minimum cross-entropy equals conditional entropy). *Let $Y \in \{1,\dots,M\}$ and $X$ be any random variable. Then*

$$\inf_{q(\cdot|X)} \mathbb{E}\big[-\log q(Y\mid X)\big] = H(Y\mid X), \tag{32}$$

*and the infimum is achieved by $q(y\mid X) = \mathbb{P}(Y = y\mid X)$.*

*Proof.* For any conditional distribution $q(\cdot\mid X)$,

$$\mathbb{E}[-\log q(Y\mid X)] = H(Y\mid X) + \mathbb{E}\big[\mathrm{KL}(p(\cdot\mid X)\,\|\,q(\cdot\mid X))\big] \geq H(Y\mid X),$$

with equality iff $q(\cdot\mid X) = p(\cdot\mid X)$ almost surely. $\qquad\square$

**Assumption .6** (Realizability by Longformer + *cls* + MLP (model-aligned)). Fix the encoder $f_\phi$. Consider the RC classifier pipeline in the main text: $\tau_{K+1}^p \xrightarrow{g_\psi} c \xrightarrow{h_{\omega_c}^{\mathrm{cls}}} \hat{\chi} \xrightarrow{\mathrm{softmax}} q_{\omega_c}(\cdot\mid c)$. Assume this model class is expressive enough such that, for the chosen summary $z_{\phi,K}$ in (21), there exist parameters $(\psi, \omega_c)$ satisfying

$$q_{\omega_c}(y\mid c) = \mathbb{P}(Y = y\mid z_{\phi,K}), \quad \forall y \in \{1,\dots,M\}, \tag{33}$$

where $c$ is computed from the sampled token sequence (and can represent any function of $z_{\phi,K}$ with sufficient capacity).

**Lemma .7** (Uniform subsampling error for mean pooling). *Assume $\|\eta(s,a)\|_2 \leq B$ for all $(s,a)$. Then for* (21)–(22)*,*

$$\mathbb{E}\Big[\big\|z_{\phi,K} - z_\phi\big\|_2^2\Big] \leq \frac{4B^2}{K}. \tag{34}$$

*Proof.* This is the standard variance bound for an empirical mean of bounded vectors. For any episode, $\|z_\phi\|_2 \leq \frac{1}{T}\sum_t \|\eta(s_t,a_t)\|_2 \leq B$. Also $\|\eta(s_{t_k},a_{t_k}) - z_\phi\|_2 \leq \|\eta(s_{t_k},a_{t_k})\|_2 + \|z_\phi\|_2 \leq 2B$. Uniform sampling gives an unbiased estimate of the mean, and thus $\mathbb{E}\|z_{\phi,K} - z_\phi\|_2^2 \leq \frac{1}{K}\mathbb{E}\|\eta(s,a) - z_\phi\|_2^2 \leq \frac{4B^2}{K}$. $\qquad\square$

## A.5 Theorem 2 (Aligned): RC Minimizes Discrepancy to the Information Goal

**RC classification loss and value function.** Define the RC classification (cross-entropy) loss:

$$\mathcal{L}_{\text{cls}}(\phi, \omega) \triangleq \mathbb{E}\big[ -\log q_{\omega_c}(Y \mid c)\big], \tag{35}$$

where $c$ is the *cls* token embedding produced by $g_\psi$ from the sampled token sequence. Define the best achievable classifier loss for a fixed encoder:

$$V_K(\phi) \triangleq \inf_\omega \mathcal{L}_{\text{cls}}(\phi, \omega), \qquad \omega_K^\star(\phi) \in \arg\inf_\omega \mathcal{L}_{\text{cls}}(\phi, \omega). \tag{36}$$

By Lemma .5 and Assumption .6, we have

$$V_K(\phi) = H(Y \mid z_{\phi,K}) = \mathcal{F}_K(\phi). \tag{37}$$

**Assumption .8** (Predictor training/estimation error). At evaluation time, the learned predictor parameters $\hat{\omega}$ satisfy

$$\mathcal{L}_{\text{cls}}(\phi, \hat{\omega}) - V_K(\phi) \le \epsilon^2. \tag{38}$$

**Assumption .9** (Cross-gradient Lipschitz and local error bound in $\omega$). Assume there exists $L_{\phi\omega} > 0$ such that

$$\|\nabla_\phi \mathcal{L}_{\text{cls}}(\phi, \omega) - \nabla_\phi \mathcal{L}_{\text{cls}}(\phi, \omega')\|_2 \le L_{\phi\omega}\|\omega - \omega'\|_2. \tag{39}$$

Moreover, assume a local quadratic-growth / error-bound condition: there exists $\mu_\omega > 0$ such that

$$\|\hat{\omega} - \omega_K^\star(\phi)\|_2^2 \le \frac{2}{\mu_\omega}\Big(\mathcal{L}_{\text{cls}}(\phi, \hat{\omega}) - V_K(\phi)\Big). \tag{40}$$

**Assumption .10** (Stochastic gradient noise). Let $\hat{g}_{\text{RC}}$ be an unbiased stochastic gradient estimator of $\nabla_\phi \mathcal{L}_{\text{cls}}(\phi, \hat{\omega})$ (e.g., mini-batch gradient). Assume

$$\mathbb{E}\big[\|\hat{g}_{\text{RC}} - \nabla_\phi \mathcal{L}_{\text{cls}}(\phi, \hat{\omega})\|_2^2\big] \le V. \tag{41}$$

**Assumption .11** (Sampling stability of the information gradient). There exists $L_{\text{sam}} > 0$ such that

$$\mathbb{E}\big[\|\nabla_\phi \mathcal{F}_K(\phi) - \nabla_\phi \mathcal{F}(\phi)\|_2^2\big] \le L_{\text{sam}}^2 \,\mathbb{E}\big[\|z_{\phi,K} - z_\phi\|_2^2\big]. \tag{42}$$

**Theorem .12** (Goal alignment via RC (stable, model-aligned)). *Under Assumptions .1 and .6–.11, define the RC discrepancy to the ideal information gradient as*

$$\Delta_{\text{RC}}(\phi) \triangleq \mathbb{E}\big[\|\hat{g}_{\text{RC}} - \nabla_\phi \mathcal{F}(\phi)\|_2^2\big]. \tag{43}$$

*Then*

$$\Delta_{\text{RC}}(\phi) \le 3V + \frac{6L_{\phi\omega}^2}{\mu_\omega}\epsilon^2 + \frac{12L_{\text{sam}}^2 B^2}{K}. \tag{44}$$

*In particular,* $\Delta_{\text{RC}}(\phi) = \mathcal{O}\big(\epsilon^2 + \frac{1}{K}\big) + V$.

*Proof.* Decompose

$$\hat{g}_{\text{RC}} - \nabla_\phi \mathcal{F}(\phi) = \underbrace{\big(\hat{g}_{\text{RC}} - \nabla_\phi \mathcal{L}_{\text{cls}}(\phi, \hat{\omega})\big)}_{(A)} + \underbrace{\big(\nabla_\phi \mathcal{L}_{\text{cls}}(\phi, \hat{\omega}) - \nabla_\phi V_K(\phi)\big)}_{(B)} + \underbrace{\big(\nabla_\phi V_K(\phi) - \nabla_\phi \mathcal{F}(\phi)\big)}_{(C)}.$$

Using $\|a + b + c\|_2^2 \le 3(\|a\|_2^2 + \|b\|_2^2 + \|c\|_2^2)$ and taking expectation,

$$\Delta_{\text{RC}}(\phi) \le 3\mathbb{E}\|(A)\|_2^2 + 3\mathbb{E}\|(B)\|_2^2 + 3\mathbb{E}\|(C)\|_2^2. \tag{45}$$

**Term (A).** By Assumption .10, $\mathbb{E}\|(A)\|_2^2 \le V$.

**Term (B).** From (40) and Assumption .8,

$$\|\hat{\omega} - \omega_K^\star(\phi)\|_2^2 \le \frac{2}{\mu_\omega}\Big(\mathcal{L}_{\text{cls}}(\phi, \hat{\omega}) - V_K(\phi)\Big) \le \frac{2}{\mu_\omega}\epsilon^2.$$

By (39),

$$\|(B)\|_2^2 = \|\nabla_\phi \mathcal{L}_{\text{cls}}(\phi, \hat{\omega}) - \nabla_\phi \mathcal{L}_{\text{cls}}(\phi, \omega_K^\star(\phi))\|_2^2 \le L_{\phi\omega}^2 \|\hat{\omega} - \omega_K^\star(\phi)\|_2^2 \le \frac{2L_{\phi\omega}^2}{\mu_\omega}\epsilon^2.$$

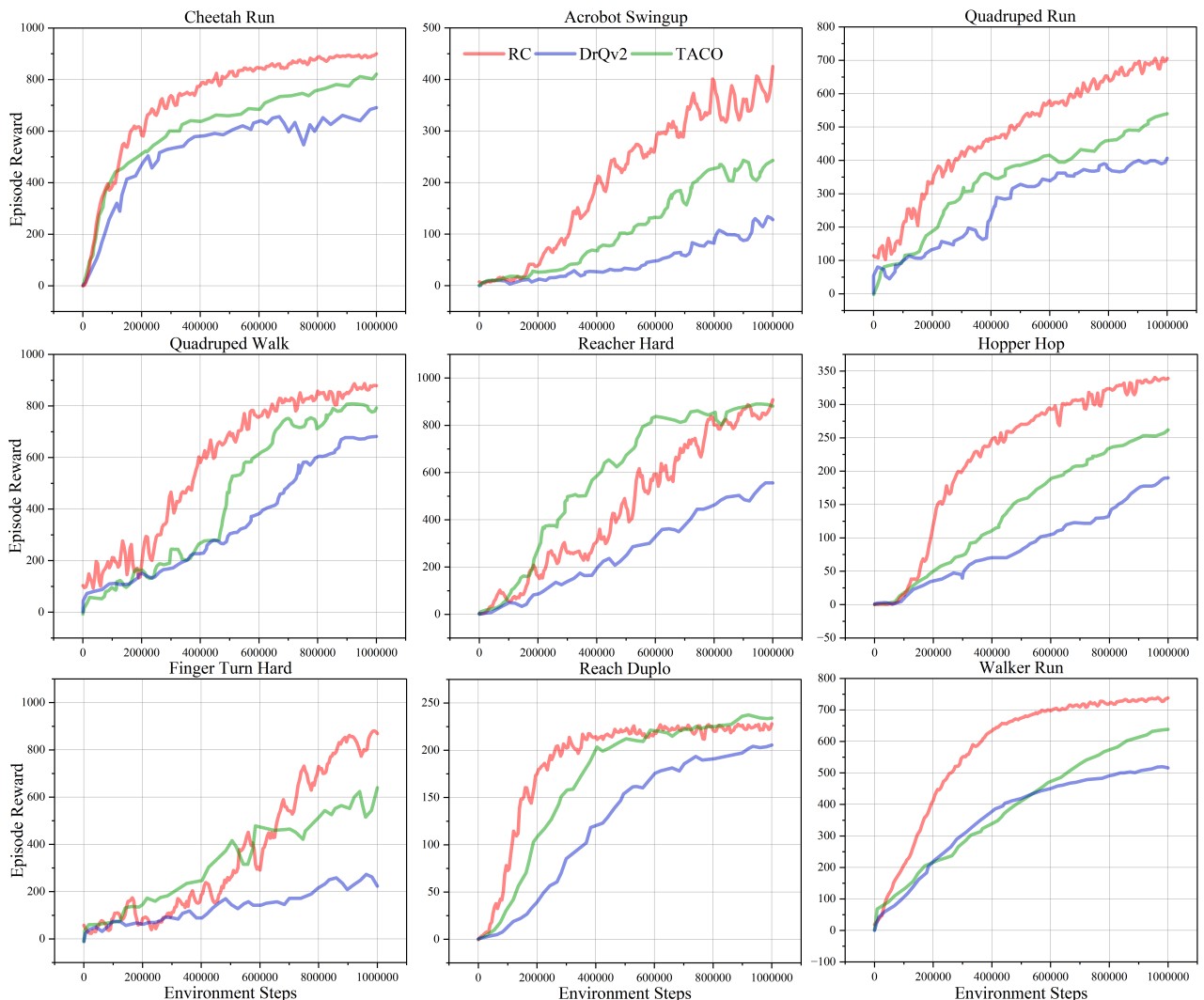

*Figure 4.* Evaluation curves of Return-Critic, TACO, and DrQv2 across nine challenging DMControl tasks.

**Term (C).** By (37), $\nabla_\phi V_K(\phi) = \nabla_\phi \mathcal{F}_K(\phi)$, thus

$$\|(C)\|_2^2 = \|\nabla_\phi \mathcal{F}_K(\phi) - \nabla_\phi \mathcal{F}(\phi)\|_2^2.$$

Assumption .11 gives

$$\mathbb{E}\|(C)\|_2^2 \le L_{\mathrm{sam}}^2 \, \mathbb{E}\|z_{\phi,K} - z_\phi\|_2^2.$$

Applying Lemma .7 yields

$$\mathbb{E}\|(C)\|_2^2 \le L_{\mathrm{sam}}^2 \cdot \frac{4B^2}{K}.$$

**Combine.** Substituting into (45) gives

$$\Delta_{\mathrm{RC}}(\phi) \le 3V + 3 \cdot \frac{2L_{\phi\omega}^2}{\mu_\omega}\epsilon^2 + 3 \cdot \frac{4L_{\mathrm{sam}}^2 B^2}{K} = 3V + \frac{6L_{\phi\omega}^2}{\mu_\omega}\epsilon^2 + \frac{12L_{\mathrm{sam}}^2 B^2}{K},$$

which is exactly (44). $\qquad\square$

**Remark (regression head and combined loss).** The proof focuses on the classification head because cross-entropy is a proper scoring rule and connects directly to conditional entropy. If one uses the combined loss $\mathcal{L}_{\text{return}} = \lambda_1 \mathcal{L}_{\text{reg}} + \lambda_2 \mathcal{L}_{\text{cls}}$, the same decomposition applies by additionally bounding the regression-gradient term, resulting in the same rate $\mathcal{O}(\epsilon^2 + 1/K) + V$ with different constants.

## Appendix B: More Experiment

### B.1 Comparison Figures in Online RL Tasks

To provide intuitive visualization of sample efficiency improvements, Figure 4 presents the evaluation curves of Return-Critic (RC), TACO (a representative auxiliary task method), and DrQv2 (our base model) across nine challenging DMControl tasks during 1 million training steps. Figure 4 clearly demonstrates RC's superior sample efficiency. In most tasks, RC achieves significantly higher returns and faster convergence rates compared to both baselines. These improvements suggest that by bridging the gap between representation and RL goals, RC is able to learn higher-quality representations that improve sample efficiency. Notably, on the Reach Duplo task, RC demonstrates faster initial convergence but eventually plateaus at a similar performance level as TACO and DrQv2. This suggests that for certain tasks where representation quality is less limiting, other factors such as exploration-exploitation balance may become the primary bottleneck.

### B.2 Attention Areas of Regressor and Classifier

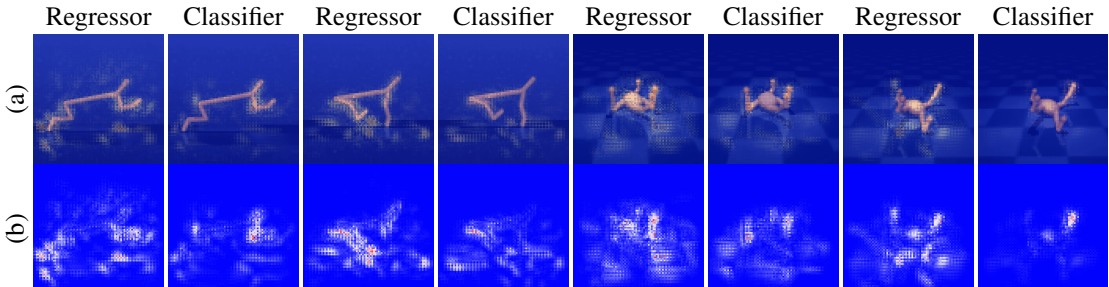

*Figure 5.* Saliency maps of regressor and classifier. (a) input images with saliency, (b) saliency maps.

To investigate the distinct roles of RC's dual prediction heads, Figure 5 presents saliency maps comparing the attention patterns of the regressor and classifier heads on Quadruped-Walk and Cheetah-Run tasks from the DMControl. As we can see, the classifier head exhibits highly focused attention on critical articulation points and active joints. This pattern confirms that the classifier learns to identify discriminative features that directly correlate with return outcomes, effectively filtering out background distractions. In contrast, the regressor head demonstrates more distributed attention patterns that capture temporal dynamics. This wider attention is essential for continuous return regression, as predicting precise return values requires understanding not only the discriminative features but also the temporal evolution of motion. These complementary attention patterns reveal why the combined $\mathcal{L}_{\text{return}}$ return outperforms either head alone. Notably, both heads avoid the irrelevant background regions, confirming our hypothesis that return prediction naturally focuses on representation aspects most relevant to long-term return.

### B.3 Attention Weight Scores for Input Sequence

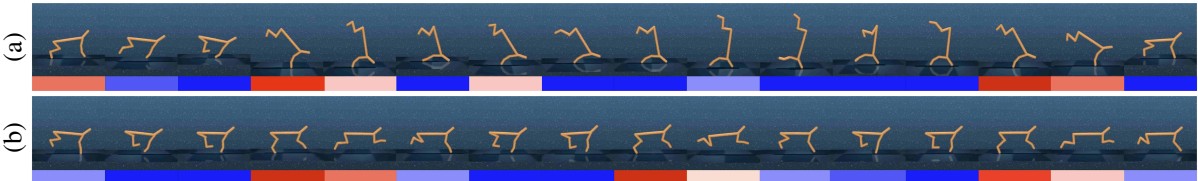

*Figure 6.* Illustrating all the single-head first attention layer weights of the *cls* token vs. 16 frames pulled from a episode. High weight values are represented by a warm color (red) while low values by a cold color (blue). (a) initial stage of training, (b) final stage of training.

To provide visual evidence of how the *cls* token dynamically allocates attention across frames within an episode, Figure 6

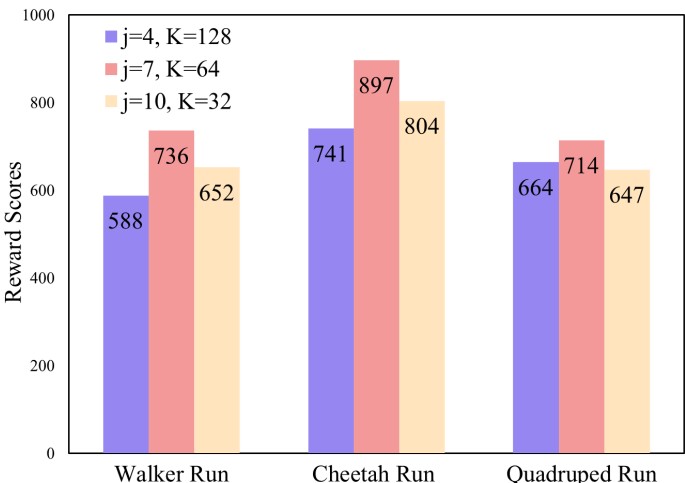

*Figure 7.* Study on uniform sampling hyperparameters of RC with 1M step size.

presents the attention weight scores for the Cheetah-Run task. The attention patterns reveal a sophisticated temporal reasoning capability that evolves throughout training. During the initial training phase, the *cls* token exhibits strong focus on frames where critical state changes occur (cheetah falling and recovering). In contrast, at the final training stage, the attention pattern shifts toward frames critical for high return (cheetah taking off). The attention patterns demonstrate that the *cls* token naturally learns to identify frames that represent important decision points in the trajectory, where state-action choices have outsized influence on return. This insight directly motivates our key function design. By leveraging these attention weights for prioritized sampling, RC implements an importance-based experience replay mechanism.

### B.4 Uniform Sampling Parameter Analysis

To optimize the uniform sampling strategy for Return-Critic (RC), we conducted a systematic analysis on the DMControl benchmark. With episode length of 1000 steps, action repeat 2, and frame stacking 3, each episode generates 500 continuous observation frames. For uniform sampling, the sampling interval $j$ and number of samples per episode $K$ must satisfy the constraint: $j \cdot (K - 1) < 500$ where $j$ is the sampling interval and $K$ is the number of frames sampled per episode. We evaluated three parameter configurations that satisfy this constraint: $j = 4, K = 128$; $j = 7, K = 64$; $j = 10, K = 32$. As shown in Figure 7, the $j = 4, K = 128$ configuration achieves 15.1% lower average performance compared to the optimal configuration. This performance degradation occurs because the small sampling interval combined with frame stacking creates highly correlated samples with excessive temporal continuity. Consequently, the return prediction task becomes too easy, reducing the effectiveness of representation learning. Conversely, the $j = 10, K = 32$ configuration shows a 10.4% performance decrease. This occurs because the large sampling interval and low sample count result in significant information loss regarding critical movement phases. This finding aligns with our theoretical that: When $K$ is too small, the sampling error becomes substantial, preventing the model from further bridging the goal discrepancy. The $j = 7, K = 64$ sampling achieves the balance between sampling error and temporal diversity.

### B.5 Longformer Parameter Analysis

To evaluate the sensitivity of Return-Critic (RC) to Longformer architecture parameters, we conducted a systematic analysis of the impact of transformer depth (number of layers) and attention head count on performance. We evaluated RC across three representative DMControl tasks (Walker Run, Cheetah Run, and Quadruped Walk) using varying Longformer configurations while keeping all other components constant. As shown in Table 6, RC demonstrates remarkable stability across different Longformer parameter configurations. The performance variance across configurations with 3, 6 layers and 1, 2, 4 attention heads remains minimal, with standard deviation below 1% across all tasks. Notably, even when using a single-layer transformer (the minimal configuration), RC maintains strong performance, with only a 3.3% average score decrease compared to the baseline configuration (3 layers, 1 heads). This insensitivity to Longformer parameters reveals a fundamental insight about the role of the transformer component in RC: it primarily serves as an effective sequence aggregator rather than a complex feature extractor. The minimal performance variation across configurations supports

our hypothesis that the performance gains of RC stem primarily from the goal-aligned return prediction objective rather than the specific architectural choice of the transformer. These findings have practical implications for RC deployment in resource-constrained settings. This parameter robustness further supports RC's practical utility, as it indicates the approach can be reliably deployed across diverse hardware platforms without extensive architecture tuning.

*Table 6.* Performance of RC with different transformer parameters.

| DMControl (1 M Step) | RC w/ 1 layer | RC w/ 3 layers | RC w/ 6 layers | RC w/ 1 head | RC w/ 2 heads | RC w/ 4 heads |
|---|---|---|---|---|---|---|
| Quadruped Run | 684±26 | 714±20 | 718±18 | 714±20 | 709±28 | 723±21 |
| Cheetah Run | 886±35 | 897±12 | 892±11 | 897±12 | 901±14 | 892±24 |
| Walker Run | 698±63 | 736±52 | 717±25 | 736±52 | 714±33 | 739±41 |
| **Average** | 756.0 | **782.3** | 775.7 | 782.3 | 774.6 | **784.6** |

## B.6 Meta-world tasks

To further demonstrate the generalizability of Return-Critic (RC) across diverse robotic manipulation domains, we evaluated its performance on the Meta-World benchmark suite. We selected three representative tasks from Meta-World: Hammer, Stick Pull, and Hand Insert. Figure 8 presents the success rates of RC compared to TACO across 1 million environment steps.

During the initial 500k training steps, both methods achieve comparable performance, with RC showing only marginal improvements. However, beyond the 500k step mark, RC demonstrates significantly accelerated learning, ultimately achieving higher final success rate on three tasks compared to TACO. This delayed but substantial improvement stems from Meta-World's inherent target randomness, where goal information vary across episodes, requiring sufficient experience to learn task-relevant features. With adequate training (beyond 500k steps), RC successfully learns to identify consistent visual patterns that correlate with task completion, enabling it to construct representations with higher return discriminability. The consistent outperformance of RC across fundamentally different task suites: DMControl, V-D4RL, and now Meta-World, demonstrates the method's robustness and general applicability, confirming that bridging goal discrepancy through return prediction provides a domain-agnostic solution to the representation learning challenge in visual reinforcement learning.

## B.7 Performance under Matched Time and Update Budgets

To explicitly isolate the true algorithmic efficiency of Return-Critic (RC) and ensure that the reported performance gains are not merely artifacts of increased computational expenditure, we provide comprehensive comparisons under strictly matched update budgets and wall-clock time constraints.

**Clarification on Matched Update-Budget.** It is important to note that our standard 1M-step experiments presented in the main text are inherently matched in terms of update budgets. In our experimental setup, the gradient update frequency is directly tied to the environment steps. Specifically, DrQv2, TACO, and RC all utilize an identical update frequency of 2 (i.e., one gradient update is performed every two environment steps). Therefore, under the standard 1M-step setting, all three methods execute exactly 500K gradient updates. Under this strictly identical budget of 500K updates, RC consistently achieves significant performance improvements over both the DrQv2 baseline and the state-of-the-art TACO method.

**Performance under Matched Wall-Clock Time.** Evaluating under strict wall-clock time limits explicitly reveals the practical efficiency of the algorithms. We conducted supplementary experiments restricting DrQv2, TACO, and RC to strict 6-hour and 12-hour wall-clock time budgets across nine challenging DMControl environments. Because DrQv2 is a lightweight baseline without auxiliary representation learning tasks, it completes approximately 1M steps in 6 hours and 2M steps in 12 hours. In contrast, RC and TACO complete roughly 0.5M steps in 6 hours and 1M steps in 12 hours.

Table 7 consolidates the performance comparisons under both the 6-hour and 12-hour limits. The results clearly demonstrate the robust superiority of RC across both time constraints:

- **Under the 6-Hour Limit:** Although RC (∼0.5M steps) has not yet reached its convergence potential within this shorter timeframe, it still achieves the best performance in the majority of tasks. Strikingly, despite executing only half the environment interactions and gradient updates of DrQv2 (∼1M steps), RC yields a significantly higher average

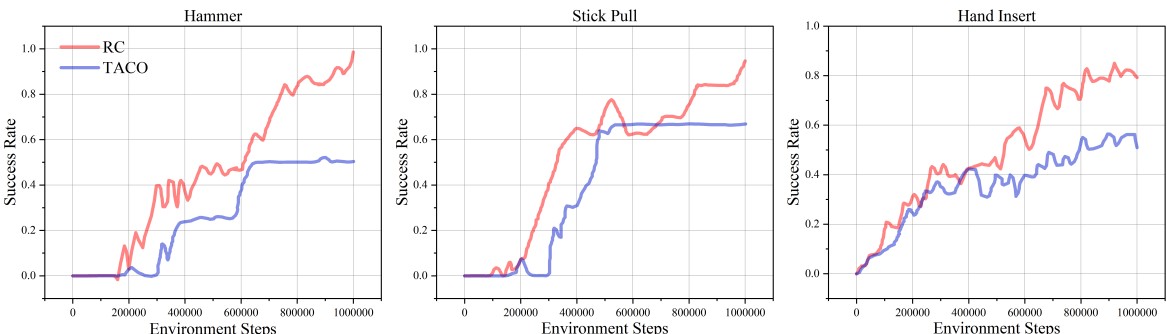

*Figure 8.* Evaluation curves of Return-Critic and TACO across three Meta-world tasks.

score (469.1 vs. 401.1).

- **Under the 12-Hour Limit:** Given a longer timeframe, DrQv2 (∼2M steps) fully plateaus at its asymptotic performance limit and eventually surpasses TACO. In stark contrast, RC (∼1M steps) maintains an 18% average performance advantage (673.9 vs. 572.6) over the fully converged DrQv2.

These results indicate that the baseline methods cannot bridge the performance gap with RC simply by extending the training time, definitively validating the inherent representation learning efficiency of the RC framework.

*Table 7.* Performance comparison under strictly equal **6-Hour** and **12-Hour** wall-clock time limits. Due to differences in algorithmic computational overhead, DrQv2 completes approximately 1M/2M steps in 6/12 hours, whereas TACO and RC complete approximately 0.5M/1M steps in the same timeframes.

| ENVIRONMENT | 6-HOUR LIMIT | | | 12-HOUR LIMIT | | |
|---|---|---|---|---|---|---|
| | DRQV2 (∼1M) | TACO (∼0.5M) | DRQV2 W/ RC (∼0.5M) | DRQV2 (∼2M) | TACO (∼1M) | DRQV2 W/ RC (∼1M) |
| QUADRUPED RUN | 407±21 | 392±27 | **523±36** | 494±7 | 541±38 | **714±52** |
| HOPPER HOP | 189±35 | 154±41 | **270±48** | 247±25 | 261±52 | **340±43** |
| WALKER RUN | 517±43 | 427±49 | **677±24** | 629±31 | 637±11 | **736±14** |
| QUADRUPED WALK | **680±52** | 452±96 | 648±63 | 867±12 | 793±8 | **875±15** |
| CHEETAH RUN | 691±42 | 644±34 | **817±22** | 822±10 | 821±4 | **897±12** |
| FINGER TURN HARD | 220±21 | **411±57** | 354±69 | 653±64 | 632±7 | **913±55** |
| ACROBOT SWINGUP | 128±8 | 102±15 | **238±46** | 323±21 | 241±21 | **450±70** |
| REACHER HARD | 572±51 | **623±62** | 483±55 | 898±14 | 883±63 | **913±45** |
| REACH DUPLO | 206±32 | 202±11 | **212±8** | 220±7 | **234±21** | 228±7 |
| AVERAGE SCORE | 401.1 | 378.6 | **469.1** | 572.6 | 560.3 | **673.9** |

## B.8 Universality in State-based Environments

While RC is primarily designed to tackle the representation learning bottleneck in visual reinforcement learning, its derived Key Score Sampling (KSS) mechanism functions as a universal module applicable to diverse off-policy algorithms regardless of the input modality. To comprehensively demonstrate the broader applicability of our framework beyond visual domains, we evaluated our method in the standard suite of MuJoCo continuous control tasks based on fully observable low-dimensional vector states.

**Experimental Setup.** To deploy RC in these state-based environments, we directly concatenate the low-dimensional states and actions to form $(s, a)$ sequences. These sequences are then fed into our Transformer-based return predictor to compute the KSS weights. For a rigorous and fair comparison, we integrated our KSS mechanism with the baseline TD3, keeping all other hyperparameters and network configurations strictly identical to the standard TD3 and Loss-Adjusted Prioritized (LAP) implementations.

**Empirical Results and Analysis.** Table 8 presents the average performance over the last 10 evaluations and 10 trials across various algorithms. Incorporating our KSS mechanism into TD3 achieves the highest performance across all evaluated challenging continuous control tasks compared to strong baselines, including SAC and TD3 equipped with other prioritized sampling methods (PER, LAP, PAL). This empirically validates two core advantages of our approach:

- **Return-Relevant Focus:** Unlike traditional TD-error-based sampling methods that might struggle to determine the true importance of transitions, the KSS mechanism explicitly selects samples that possess a direct causal relationship with the final episodic return. It accurately captures both the critical positive $(s, a)$ pairs leading to high returns and the critical erroneous $(s, a)$ pairs leading to low returns.

- **Robustness Against Outliers:** Previous studies indicate that prioritizing samples with Mean Squared Error (MSE) can introduce severe bias by favoring outliers. Our hybrid sampling approach naturally mitigates this issue. By applying KSS to exactly half of the sampled batch and utilizing standard uniform sampling for the remaining half, we successfully prevent the value network from being biased towards uninformative outliers, ensuring stable and robust learning.

*Table 8.* Average performance over the last 10 evaluations and 10 trials on State-based MuJoCo (3M Steps). Integrating KSS significantly improves the performance of the baseline TD3 and outperforms other traditional sampling strategies.

| ENVIRONMENT | TD3 | SAC | TD3 + PER | TD3 + LAP | TD3 + PAL | TD3 + KSS (OURS) |
|---|---|---|---|---|---|---|
| HALFCHEETAH | 13570.9±794.2 | 15511.6±305.2 | 13927.8±683.9 | 14836.5±532.2 | 15012.2±885.4 | **15846.8±563.2** |
| HOPPER | 3393.2±381.9 | 2851.6±417.4 | 3275.5±451.8 | 3246.9±463.4 | 3129.1±473.5 | **3597.1±453.5** |
| WALKER2D | 4692.4±423.6 | 5234.4±346.1 | 4719.1±492.0 | 5230.5±368.2 | 5218.7±422.6 | **6138.4±389.3** |
| ANT | 6469.9±200.3 | 4923.6±882.3 | 6278.7±311.3 | 6912.6±234.4 | 6476.2±640.2 | **7563.1±529.2** |
| HUMANOID | 6437.5±349.3 | 6580.9±296.6 | 5629.3±174.4 | 7855.6±705.9 | 8265.9±519.0 | **9237.9±685.2** |
| AVERAGE SCORE | 6912.8 | 7000.4 | 6756.1 | 7616.4 | 7620.4 | **8476.7** |

## Appendix C: Implementation Details

### C.1 Pseudocode

Pseudocode 1.

### C.2 Computing Hardware

All the experiments were run on a desktop machine (Intel i7, 14th generation processor, 64GB RAM) with a single NVIDIA RTX 4090 GPU. We report the training wall time and VRAM consumption of different methods on DMControl tasks in Table 9.

*Table 9.* Wall time and VRAM consumption comparison of different methods on DMControl tasks.

| Algorithm | Wall Time (1M Steps) | VRAM |
|---|---|---|
| CURL | $\sim$ 8 hours | $\sim$ 3.7 GB |
| DrQv2 | $\sim$ 6 hours | $\sim$ 1.7 GB |
| PlayVirtual | $\sim$ 18 hours | $\sim$ 16.2 GB |
| MLR | $\sim$ 20 hours | $>$ 24.0 GB |
| TACO | $\sim$ 12 hours | $\sim$ 5.1 GB |
| MIND | $\sim$ 24 hours | $\sim$ 18.0 GB |
| **DrQv2 w/ RC (Ours)** | $\sim$ **12 hours** | $\sim$ **2.6 GB** |

### C.3 Hyperparameters

Return-Critic introduces additional hyperparameters to DrQv2 (Yarats et al., 2022): sampling interval $j$ and number of samples per episode $K$, number of transformer layers and attention heads, weights $\lambda_1$ and $\lambda_2$ used to balance the weight of regressor and classifier loss, and temperature $\mu$ of sampling probability. Table 10 summarizes the hyperparameters used in all experiments.

---

**Algorithm 1** Pseudocode of Return-Critic (RC)

---

**Parameter**: frequency of RC updates $N_{\text{aux}}$, learning rate $\alpha$, network parameters $\theta$ (including visual encoder, actor, critic, and RC components), replay buffer $\mathcal{B}$

  Initialize network parameters $\theta$, replay buffer $\mathcal{B}$
  **for** each interaction time step **do**
    Take action $a \sim \pi_\theta(\cdot|s)$, observe reward $r$, next state $s'$
    Store transition $(s, a, r, s')$ in $\mathcal{B}$
    **if** enough samples in $\mathcal{B}$ **then**
      Sample mini-batch $\{(s_i, a_i, r_i, s'_i)\}_{i=1}^N$ from $\mathcal{B}$
      Update critic: $\theta \leftarrow \theta - \alpha\nabla_\theta\mathcal{L}_Q(\theta)$
      Update actor: $\theta \leftarrow \theta - \alpha\nabla_\theta\mathcal{L}_\pi(\theta)$
      **if** step mod $N_{\text{aux}} = 0$ **then**
        Sample $\hat{T}$ episodes from $\mathcal{B}$
        Initialize prioritized batch $\mathcal{B}_{\text{pri}} \leftarrow \emptyset$
        Uniformly sample $K$ samples from episodes with interval $j$
        Encode samples to state representations $s_t$ using visual encoder $f_\theta$
        Form state-action tokens and process through Longformer to get *cls* token $c$
        Compute return prediction loss: $\mathcal{L}_{\text{return}} = \lambda_1\mathcal{L}_{\text{reg}} + \lambda_2\mathcal{L}_{\text{cls}}$
        Compute key function loss: $\mathcal{L}_{\text{key}} = \text{MSE}(\alpha, \kappa_\theta(s, a))$ using attention weights
        Add $\{(s_i, a_i, r_i, s'_i)\}_{i=1}^{\hat{T}\cdot K}$ pairs to $\mathcal{B}_{\text{pri}}$ with key scores $\kappa_\theta(s, a)$
        Sample prioritized mini-batch from $\mathcal{B}_{\text{pri}}$ according to sampling probability $P(i)$
        Update critic with prioritized samples: $\theta \leftarrow \theta - \alpha\nabla_\theta\mathcal{L}'_Q(\theta)$
        Update parameters: $\theta \leftarrow \theta - \alpha\nabla_\theta(\mathcal{L}_{\text{return}} + \mathcal{L}_{\text{key}})$
      **end if**
    **end if**
  **end for**

---

*Table 10.* Hyperparameters.

| Hyperparameter | Value |
| --- | --- |
| Frame rendering | $84 \times 84 \times 3$ |
| Stacked frames | 3 |
| Action repeat | 2 |
| Discount factor $\gamma$ | 0.99 |
| Seed frames | 4000 |
| Exploration steps | 2000 |
| Mini-batch size | 256 |
| Replay buffer size | $10^6$ |
| Optimizer | Adam |
| Learning rate | $10^{-4}$ |
| Agent update frequency | 2 |
| Sampling interval $j$ | 7 |
| Samples of per episode $K$ | 64 |
| Transformer layers | 3 |
| Attention heads | 1 |
| Number of episodes $\hat{T}$ | 8 |
| Weights $\lambda_1$ and $\lambda_2$ | 0.5 and 0.5 |
| Sampling probability temperature $\mu$ | 0.1 |
| Features dim. | 50 |
| Hidden dim. | 1024 |

