# OpenReview forum: "Return-Critic: Bridging Goal Discrepancy for Efficient Visual Reinforcement Learning"
_ICML.cc/2026/Conference — ICML 2026 regular_

### Official Review · Reviewer_4W1q · 2026-02-25

**Soundness:** 3
**Presentation:** 3
**Significance:** 3
**Originality:** 3
**Overall Recommendation:** 4
**Confidence:** 4

**Summary:**

This paper proposes Return-Critic (RC), an approach that introduces an auxiliary representation learning loss based on episodic return prediction. By utilizing a Transformer to predict the return of an episode, the method forces the visual encoder to extract representations that are explicitly relevant to the return. Furthermore, the attention weights generated by the Transformer are used to compute "key scores," which subsequently dictate the priority of transitions for sampling from the replay buffer. The authors evaluate RC on both online (Visual DMC) and offline (V-D4RL) visual reinforcement learning benchmarks, demonstrating improvements over prior state-of-the-art baselines.

**Compliance With Llm Reviewing Policy:**

Affirmed.

**Final Justification:**

I believe this work proposes an interesting method supported by extensive experiments. The authors have also provided additional state-based experiments during the rebuttal to demonstrate the effectiveness of the KSS mechanism. My concerns have been addressed.

**Key Questions For Authors:**

1. Could the authors provide an ablation study that completely isolates the representation learning component from the key score sampling? Specifically, how does RC perform if the replay buffer utilizes standard uniform sampling or other sampling methods?
2. While the primary focus is on visual RL (where representation learning is very difficult), the RC framework theoretically applies to any MDP. Would it be possible to provide some results on state-based environments (e.g., MuJoCo)?

**Limitations:**

yes

**Strengths And Weaknesses:**

**Strengths**

1. The main idea of tying representation learning directly to episodic returns via a Transformer is intuitive, and the inclusion of supporting theoretical analysis strengthens the underlying motivation.
2. The experimental validation is extensive, covering both online and offline visual RL domains against competitive baselines.
3. The authors have provided the source code, which facilitates verification of their empirical claims.

**Weaknesses**

1. To isolate the efficacy of the key score-based priority sampling, the authors should compare it against standard uniform sampling, as well as established prioritized sampling techniques such as Prioritized Experience Replay (PER) [1] and Loss-Adjusted Prioritized (LAP) [2].

   [1] Tom Schaul, John Quan, Ioannis Antonoglou, David Silver. Prioritized Experience Replay. ICLR 2016.

   [2] Scott Fujimoto, David Meger, and Doina Precup. LAP: An Equivalence between Loss Functions and Non-Uniform Sampling in Experience Replay. NeurIPS 2020.

2. The authors utilize fixed hyperparameters ($\lambda_1=0.5$ and $\lambda_2=0.5$) across all environments. While finding a universal hyperparameter is beneficial, the lack of a sensitivity analysis leaves it ambiguous how robust the RC framework is to these values.

---

> ### Author Rebuttal · Authors · 2026-03-29
>
> Thank you for your high evaluation of the intuitiveness of the core ideas of this article and the comprehensiveness of the experiments. We have carefully studied your constructive suggestions and have prepared the corresponding experimental results and analysis.
>
> Response to **Question 1** and **Weakness 1**.
>
> This is a highly insightful question. You correctly point out that the original manuscript lacked an analysis of **Key Score Sampling (KSS)** as an independent plug-in applied to the baseline algorithm (e.g., DrQv2). In Table 4 of the original paper, the w/o L_key variant explicitly disables prioritized sampling (reverting to uniform sampling), resulting in an approximate 9% performance drop compared to the full RC. This successfully isolates and confirms the specific contribution of KSS.
>
> To comprehensively address your concern, we have conducted extensive cross-ablation experiments, evaluating both RC and DrQv2 across various sampling strategies (Uniform, KSS, PER, LAP, PAL). 1) Empirical Results: Both RC and DrQv2 equipped with KSS achieved a significant performance boost of approximately 10%. In contrast, LAP and PAL yielded only about a 4% improvement, and the traditional PER showed no significant performance gain. 2) Mechanistic Differences and Intuition: Why does KSS outperform traditional methods? PER, LAP, and PAL rely on single-step TD-errors (i.e., how poorly a (s, a) is learned) to determine sampling rates. However, a high TD-error does not necessarily imply that the (s, a) is important for the final return (in continuous control tasks, the (s, a) pairs are dense, and most (s, a) have little impact on the return). Oversampling these uninformative, high-error (s, a) easily leads to overfitting to irrelevant (s, a). Conversely, our KSS is based on the Transformer's Credit Assignment mechanism. It determines sampling rates based on the "criticality" of a (s,a) pair to the overall return, accurately focusing on key (s,a) that directly impact the final outcome: capturing both positive actions (e.g., jumping) and critical mistakes (e.g., falling). 3) Implementation Detail: It is important to note that, in practice, KSS is only applied to 50% of the sampled batch, while the remaining 50% uses standard uniform sampling. As demonstrated in Figure 6, critical (s,a) pairs are sparse. This hybrid approach prevents the value network's MSE loss from being biased towards outliers.
>
> **Table**: Return scores of RC and DrQv2 equipped with different sampling strategies (KSS, PER, LAP, PAL) at 1M steps. The best results are highlighted in bold.
>
> | DMControl (1M Step) | RC w/ KSS | RC w/ PER | RC w/ LAP | RC w/ PAL | DrQv2 w/ KSS | DrQv2 w/ PER | DrQv2 w/ LAP | DrQv2 w/ PAL |
> | :--- | :---: | :---: | :---: | :---: | :---: | :---: | :---: | :---: |
> | Walker Run | **714±20** | 646±20 | 687±15 | 677±16  | **580±20** |  509±22 | 537±15 | 542±27 |
> | Cheetah Run | **897±12** | 822±14 | 843±11 | 845±14 | **764±14** |  678±28 | 701±17 | 696±21 |
> | Quadruped Run | **736±52** |  658±21 | 679±18 | 692±23 | **457±19** |  423±15 | 443±21 | 437±11 |
> | **Average** | **782.3** | 708.7 | 736.3 | 738.0 | **600.3** |  536.6 | 560.3 | 558.3 |
>
> Response to **Question 2**.
>
> This is an extremely insightful and forward-looking question. The short answer is: yes. While RC is primarily designed to tackle the representation learning bottleneck in Visual RL, its derived Key Score Sampling (KSS) mechanism is a universal module that can be applied to any off-policy RL algorithm, regardless of the input modality. As demonstrated in our supplementary experiments above, even when we apply a stop-gradient operation to the state representations (relying solely on KSS to prioritize the replay buffer) the model still achieves a significant performance improvement over the baseline. This strongly proves the broad potential of RC's credit assignment and the universality of the KSS mechanism. Therefore, we firmly believe that applying KSS to fully observable, low-dimensional vector states (state-based MuJoCo) will yield highly effective improvements. Because our current research infrastructure and computational pipelines are heavily tailored to Visual RL, we regret that we could not complete a rigorous and comprehensive set of state-based experiments within the short time limit of the rebuttal phase. However, we fully commit to adding these state-based experiments and corresponding analyses in the final revision to thoroughly demonstrate the broader applicability of our framework beyond visual domains.
>
> Response to **Weakness 2**.
>
> While using fixed hyperparameters across all diverse environments without per-task tuning already indicates RC's strong inherent robustness, we agree that a systematic sensitivity analysis is valuable. In the revised appendix, we will add comprehensive experiments detailing how varying these weights impacts sample efficiency and final performance.

---

> > ### Author Rebuttal · Reviewer_4W1q · 2026-04-02
> >
> > My concerns have been largely addressed. I acknowledge that a comprehensive study on state-based environments may not be feasible within the short rebuttal period, and I look forward to seeing such results in a future version of the manuscript.

---

> > > ### Author Response · Authors · 2026-04-05
> > >
> > > Dear Reviewer 4W1q,
> > >
> > > Thank you so much for your kind understanding, your recognition of our rebuttal, and for acknowledging that your concerns have been largely addressed.
> > >
> > > Your comment about looking forward to the state-based experiments in a future version deeply motivated us. We completely agree that demonstrating the universality of our framework on standard low-dimensional states is a crucial piece of the puzzle. Therefore, to ensure this paper is as rigorous and comprehensive as possible, we prioritized these state-based evaluations and have successfully completed them within the rebuttal window.
> > >
> > > Experimental Setup:
> > > To ensure a rigorous and direct comparison, we evaluated our method in the standard suite of MuJoCo continuous control tasks, exactly consistent with the environments used in the LAP/PAL paper. To deploy RC in these state-based environments, following the uniform sampling strategy within the same episode used in the original RC, we directly concatenate the low-dimensional states and actions to form (s, a) sequences. These sequences are then fed into our Transformer-based return predictor to compute the Key Score Sampling (KSS) weights. For a fair comparison, we integrated our KSS mechanism with the baseline TD3, keeping all other hyperparameters and network configurations strictly identical to the standard TD3 and LAP implementations.
> > >
> > > The table below presents our results alongside the benchmark scores from Table 1 of the LAP paper.
> > >
> > > **Table: Average performance over the last 10 evaluations and 10 trials on State-based MuJoCo.**
> > >
> > > | Environment (3M Steps) | TD3 | SAC | TD3 + PER | TD3 + LAP | TD3 + PAL | TD3 + KSS |
> > > | :--- | :---: | :---: | :---: | :---: | :---: | :---: |
> > > | HalfCheetah | 13570.9±794.2 | 15511.6±305.2 | 13927.8±683.9 | 14836.5±532.2 | 15012.2±885.4 | **15846.8±563.2** |
> > > | Hopper | 3393.2±381.9 | 2851.6±417.4 | 3275.5±451.8 | 3246.9±463.4 | 3129.1±473.5 | **3597.1±453.5** |
> > > | Walker2d | 4692.4±423.6 | 5234.4±346.1 | 4719.1±492.0 | 5230.5±368.2 | 5218.7±422.6 | **6138.4±389.3** |
> > > | Ant | 6469.9±200.3 | 4923.6±882.3 | 6278.7±311.3 | 6912.6±234.4 | 6476.2±640.2 | **7563.1±529.2** |
> > > | Humanoid | 6437.5±349.3 | 6580.9±296.6 | 5629.3±174.4 | 7855.6±705.9 | 8265.9±519.0 | **9237.9±685.2** |
> > > | **Average Score** | 6912.8 | 7000.4 | 6756.1 | 7616.4 | 7620.4 | **8476.7** |
> > >
> > > As demonstrated in the table, incorporating our KSS mechanism (derived from RC) into TD3 achieves the highest performance across these challenging continuous control tasks. This empirically validates two core advantages of our approach: **(1) Return-Relevant Focus:** Unlike TD-error-based sampling methods (like PER or LAP) which might struggle with determining true importance, our KSS mechanism explicitly selects samples that have a direct causal relationship with the final episodic return. It accurately captures both the critical positive (s, a) pairs (leading to high returns) and the critical erroneous (s, a) pairs (leading to low returns). **(2) Robustness Against Outliers:** Previous studies note that prioritizing samples with Mean Squared Error (MSE) can introduce severe bias by favoring outliers. Our method naturally mitigates this issue. By applying KSS to only 50% of the sampled batch and utilizing standard uniform sampling for the remaining 50%, we successfully prevent the value network's MSE loss from being biased towards uninformative outliers, ensuring stable and robust learning.
> > >
> > > **Conclusion:**
> > >
> > > We noticed that the acknowledgement status was marked as (c) Partially resolved, due to the difficulty of addressing it in a short rebuttal. We hope that by delivering these state-based results, we have now fully resolved this final remaining concern.
> > >
> > > We are deeply grateful for your forward-looking question, which explicitly expanded the universality and impact of our work. We hope these newly added empirical results provide you with full confidence in our method, and we would be deeply grateful if they might encourage you to update the acknowledgement status and view our submission favorably. Thank you once again for your invaluable guidance!

---

### Official Review · Reviewer_NXir · 2026-03-08

**Soundness:** 3
**Presentation:** 2
**Significance:** 2
**Originality:** 3
**Overall Recommendation:** 3
**Confidence:** 3

**Summary:**

This paper proposes Return-Critic (RC) to improve representation learning for visual reinforcement learning. The goal of RC is to learn to predict the return and the importance of sequential observations, such that the key features of the raw observation are encoded. A Longformer Encoder is used as a backbone to learn the representation, and the RC framework works as an auxiliary objective for both online RL methods and offline RL methods. The authors provide theoretical analysis to show that representation with goal alignment improves representation quality. Experiments in DMControl and V-D4RL tasks show that the method can improve existing methods and outperform other visual RL representation learning methods.

**Compliance With Llm Reviewing Policy:**

Affirmed.

**Final Justification:**

This is an interesting paper with novel ideas. The authors have addressed some of my concerns, although not all. To better justify the claims of the author (e.g. the source of the suboptimality is the base algorithm), it may be needed to present more experiment results across various base methods and various data/environment setups.
After reading other reviewer's comments and the author responses, I maintain my current rating.

**Key Questions For Authors:**

1. Can the authors show more details on computational cost comparison in experiments?
2. Can the authors give more explanation of importance prediction and training stability?
3. How can one overcome suboptimal data coverage?

**Limitations:**

Yes

**Strengths And Weaknesses:**

Strengths:
- The method can serve as au auxiliary framework and combined with existing RL algorithms to improve their representation learning.
- The method works in both offline and online scenarios.
- The idea of importance prediction sounds novel to me.
- Both theoretical results and empirical results are solid.

Weaknesses:
- The proposed method requires additional transformer training with long sequence, while it lacks detailed analysis on the computational efficiency. Can the authors provide more analysis and experiments to compare computation efficiency? such as resources, time, etc.
- In Table 3, it seems that in the expert scenario, the RC method is not as good as BC or CQL. Does it suggests that the algorithm depends on the diversity or coverage of offline dataset? In online dataset where the data is generated by a policy, how can we prevent the algorithm from learning suboptimal representations from limited observations when exploration is not sufficient?
- The explanation of the algorithm, especially the importance prediction part, is not very clear. Could the authors explain more of the intuition behine the design? Since $\alpha$ is also a learned variable, how can one stably learn Eq (7)?

---

> ### Author Rebuttal · Authors · 2026-03-29
>
> Thank you for recognizing the motivation and experimental results of RC. We have addressed your questions one by one below.
>
> Response to **Question 1**.
>
> Thank you for your attention to the details of computational overhead. First, regarding time efficiency (detailed in Appendix Table 6), although RC takes longer to run 1M steps in the DMControl environment (about 12 hours) compared to the base algorithm DrQv2 (about 6 hours), this computational investment yields a massive 68% improvement in average scores (as shown in Figure 4, with improvements exceeding 100% on some challenging tasks). More importantly, in a fair comparison with similar SOTA auxiliary task methods, RC's runtime is comparable to TACO (about 12 hours), yet it achieves a significant 20% lead in average task scores. Second, regarding model computational complexity, the Longformer used in RC features linear complexity, and we employ an extremely minimalistic Transformer configuration of only "a single head and three layers." Combined with the parameter sensitivity analysis in Appendix Table 5, this demonstrates that RC's additional demand for computational resources is remarkably low; its success relies primarily on the architectural design of the "return prediction" alignment objective, rather than simply relying on computing power or stacking parameters. Finally, we conducted empirical measurements of peak VRAM usage under the exact same hardware environment: RC occupies only 2.6GB of VRAM. While slightly higher than DrQv2 (1.7GB), it is more memory-efficient than TACO (5.1GB) and drastically lower than heavy baselines like MIND (18.0GB), PlayVirtual (16.2GB) and MLR (over 24GB). This comprehensively proves that RC possesses high cost-effectiveness and significant advantages in terms of resource consumption and computational budget. More details will be provided in the appendix.
>
> Response to **Question 2**.
>
> Thank you for your in-depth discussion regarding the importance prediction (Eq. 7) and its stability.
>
> Regarding the Importance Prediction: The design of importance prediction is based on the inherent Credit Assignment capability of the Transformer. The Transformer's credit assignment mechanism stems from the attention mechanism's ability to automatically focus on discriminative inputs; it highlights the inputs contributing to high returns and those leading to low returns, allowing us to leverage this capability to optimize the sampling process. Figure 6 clearly demonstrates that as training progresses, attention weights automatically focus on key frames that have a decisive impact on the final return (such as the moment of jumping or falling). Recent work, AttnRL [1] has also validated this automatic focusing capability of the attention mechanism. Therefore, extracting the normalized attention weight score $\alpha$ as a pseudo-label to measure sample importance is theoretically intuitive and empirically effective.
>
> Regarding Training Stability:  $\alpha$ is obtained by normalizing the attention weights. Because the Transformer possesses the aforementioned Credit Assignment mechanism, the attention distribution naturally stabilizes as the auxiliary task converges. Empirically, RC can predict returns accurately within a short span of 10K steps, meaning $\alpha$ becomes stable very early in training. Furthermore, when calculating the loss for Eq. (7), we apply a detach (stop-gradient) operation to the target score $\alpha$, which cuts off gradient backpropagation and prevents it from disrupting the encoder. Concurrently, during the backpropagation of the importance prediction loss, we additionally introduce strict Gradient Clipping, eliminating the risk of potential numerical overflow or exploding gradients. The rapidly stabilizing $\alpha$ combined with these designs collectively guarantees the stable training of the Eq. (7).
>
> Response to **Question 3**.
>
> While purely offline Expert datasets lack the return variance RC needs, we alleviate suboptimal data coverage in standard online visual RL through three synergistic mechanisms. First, the inherent exploration noise of online algorithms continuously generates a mix of diverse trajectories, providing the crucial "return variance" required for RC to distinguish key features. Second, our massive replay buffer (1,000,000 transitions) securely retains these diverse, early exploratory experiences. Finally, our Key-score Prioritized Sampling actively increases the exposure of highly critical states. Together, these mechanisms ensure that the encoder avoids overfitting to frequent but uninformative states. By actively filtering for the critical states that actually drive the variance in returns, RC extracts robust, return-related representations, reducing the impact of the overall data distribution skew.
>
> [1] Runze Liu, Jiakang Wang, Yuling Shi et al. Attention as a Compass: Efficient Exploration for Process-Supervised RL in Reasoning Models. ICLR 2026

---

> > ### Author Rebuttal · Reviewer_NXir · 2026-04-06
> >
> > Thank you for the author's response. Regarding the question of expert datasets, I think what the author said justifies my concern that the algorithm relies on data coverage. However, the algorithm's performance in this scenario is suboptimal despite these mechanisms. So I will keep my original score.

---

> > > ### Author Response · Authors · 2026-04-06
> > >
> > > Dear Reviewer NXir,
> > >
> > > Thank you for your follow-up and for pinpointing the exact reason for your remaining concern. We completely understand your observation regarding the performance on the expert dataset. However, we would like to respectfully clarify the root cause of this suboptimality, as it fundamentally stems from the base algorithm rather than our proposed RC framework.
> > >
> > > **1. The source of the suboptimality is the base algorithm, not RC:**
> > >
> > > As shown in Table 3 of our manuscript, the vanilla base method we used for the offline setting (DrQv2 + BC) inherently suffers from severe performance drops on the expert and mixed datasets. This is a known limitation of using simple Behavioral Cloning (BC) constraints in certain offline coverage scenarios. Crucially, our RC method does not introduce this vulnerability. Instead, as the table demonstrates, adding RC to DrQv2 + BC actually partially alleviates this inherent problem, yielding consistent performance improvements over the vanilla base method even on these difficult datasets. While RC does not completely cure the base algorithm's flaws, it strictly improves the representation quality and final score in every dataset tested.
> > >
> > > **2. The primary scope and contribution of RC:**
> > >
> > > We respectfully emphasize that the core contribution of RC is alleviating the "goal discrepancy" problem to extract return-relevant features. Our primary target domain is online continuous Visual RL, where data coverage is naturally addressed through exploration, allowing RC to achieve a massive 20% average performance lead over the current SOTA.
> > >
> > > The offline V-D4RL experiments were included purely to demonstrate the universality of RC as a plug-and-play auxiliary module, proving that it can successfully enhance existing offline algorithms. We are not proposing a new state-of-the-art offline RL algorithm tailored to solve offline data-coverage limits.
> > >
> > > **Conclusion:**
> > >
> > > We sincerely hope this clarifies that the suboptimal performance in the expert scenario is a legacy limitation of the specific base algorithm (DrQv2 + BC), rather than a flaw in the RC representation learning framework itself. Since RC strictly improves the performance of the base algorithm across all datasets, while also dominating in its primary online setting, we hope you might consider viewing RC's core contribution more favorably.
> > >
> > > Thank you once again for your rigorous engagement and valuable time!

---

### Official Review · Reviewer_VnXb · 2026-03-13

**Soundness:** 2
**Presentation:** 2
**Significance:** 2
**Originality:** 2
**Overall Recommendation:** 3
**Confidence:** 4

**Summary:**

The paper aims to address the sample inefficiency of pixel-based visual reinforcement learning, which the authors attribute to inadequate state representation learning and a mismatch between existing auxiliary objectives and the ultimate RL objective of return maximization. To this end, they propose Return-Critic (RC), an auxiliary framework that predicts episode return from partial frames so as to encourage the visual encoder to learn return-relevant representations, and they evaluate it extensively on both online DMControl and offline V-D4RL benchmarks. The reported results show substantial gains in sample efficiency, but I hold an opposing view on the central premise, as explained below.

**Compliance With Llm Reviewing Policy:**

Affirmed.

**Final Justification:**

Thank you for the detailed response. However, as Q1, Q2, and Q3 still require important clarification in the main text or appendix, I would raise my rating to weak reject, but not to accept.

**Key Questions For Authors:**

1. In the third paragraph of the Introduction (Lines 62–67), the authors state that, in conventional off-policy visual RL, the objective of representation learning is merely to predict immediate rewards. However, in RL, the TD loss constrains the current value estimate to approach the true expected return under the Bellman equation. In visual RL, the visual encoder is trained through gradients propagated from the TD loss. Therefore, representation learning in visual RL already implicitly captures the true long-term return.

2. In the third paragraph of the Introduction, the discussion of existing work lacks several key citations.

3. The proposed RC method is only combined with data augmentation methods. Does RC conflict with existing self-supervised methods? If not, why was it not combined with them to examine how much additional performance gain can be obtained? In addition, experiments combining RC with the most basic SAC baseline are also missing.

4. For the visualizations, the paper does not provide sufficient implementation details regarding how the heatmaps are generated.

**Strengths And Weaknesses:**

The proposed method demonstrates strong empirical performance, but it seems to primarily reinforce the alignment between visual representations and future return that is already inherently present in standard visual RL. Moreover, the core comparative experiments are limited to combinations with data augmentation-based methods, which is not sufficient for a comprehensive evaluation.

---

> ### Author Rebuttal · Authors · 2026-03-28
>
> Thank you for your rigorous review. We have responded to your questions one by one below.
>
> Response to **Question 1**.
>
> This is a profound and core discussion. We acknowledge that the statements in our original manuscript were indeed not comprehensive enough, and we will supplement and revise these incomplete statements in the paper.
>
> Under an ideal MDP with state-based inputs or tabular representations, TD learning can indeed learn and gradually converge to the true expected return. However, in the deep network optimization of high-dimensional visual RL (VRL), this assumption often fails. First, the partial observability of pixel inputs, high-dimensional redundancy, inherent fitting errors of neural networks, and indirect gradient backpropagation make it impossible for pixel-based deep reinforcement learning to acquire an accurate current state. Second, TD relies solely on the difference between n-step future value predictions for gradient backpropagation, lacking an accurate global constraint on future value predictions. This makes it severely subject to the accumulation of bootstrapping errors. The difficulty in acquiring states, coupled with the accumulation of bootstrapping errors, makes it extremely difficult for TD methods in VRL to converge to the true expected return.
>
> Furthermore, we discovered that when conventional VRL methods use TD updates, the difference between their predicted current state cumulative value and the target current state cumulative value is much smaller than the difference between the predicted current state cumulative value and the true current state cumulative value. This phenomenon indicates that value prediction biases caused by inaccurate states and overestimation primarily occur at the distal end (long-term) rather than the proximal end (short-term). This shows that while TD in VRL can learn proximal immediate rewards very well, it exhibits significant bias regarding distal long-term returns. Therefore, we propose that in traditional off-policy VRL, the encoder learns representations mainly related to immediate rewards from the observations.
>
> Figure 3 also provides intuitive counter-evidence. If TD could effectively capture long-term returns, the attention regions of the Q-Critic and the Return-Critic should be similar. However, the saliency maps show that the Q-Critic only focuses on the contact points with the ground (local features for immediate rewards), whereas the Return-Critic precisely focuses on the robot's joints (the core determining the long-term trajectory). RC provides a global, true supervision signal for the full episode, effectively bypassing the bootstrapping errors of n-step TD, thereby guiding the visual encoder to learn representations related to the total return.
>
> Response to **Question 2**.
>
> We sincerely appreciate your reminder. In the third paragraph of the revised manuscript, we will carefully incorporate the latest key references concerning the limitations of representation learning in visual RL, the mechanisms of TD bootstrapping errors, and related cutting-edge auxiliary tasks.
>
> Response to **Question 3**.
>
> 1)Data augmentation: Recent advanced methods (ResAct, TACO, MLR, etc.) universally use data augmentation. The recent paper [1] points out that data augmentation aims to alleviate the network's plasticity loss rather than improving representations. This is complementary to RC's goal of "improving representation quality." 2) Combination with SSL: Mechanistically, RC does not conflict with existing SSL methods. We choose not to combine them because existing SSL methods possess "goal discrepancy," and RC aims to replace them (already outperforming TACO by 20%). Forcibly stacking them causes redundancy and contradicts our fundamental motivation. 3) Basic SAC baseline: This was our oversight. Since recent methods universally use data augmentation, we omitted the ablation study on it. We supplement the brief experimental data.
>
> |  |  RC + SAC | TACO + SAC |
> | :--- | :---: | :---: |
> | Walker Run | 608±22 | 467±34 |
> | Cheetah Run | 755±32 |  682±52 |
> | Quadruped Run | 546±28 |   424±41  |
> | Average | 636.3 | 524.3 |
>
> Response to **Question 4**.
>
> As stated in Section 4.3 of the paper, we utilized the Guided Backpropagation technique. Specifically, we obtain feature maps through the network's forward pass. During the backward pass, negative gradients are clipped to zero. Finally, we extract the gradients from the last layer of the visual encoder and multiply them with the input features to generate high-resolution heatmaps. Figure 1 and Figure 3 illustrate the differences in the regions of the observation images focused on by representations extracted using different auxiliary tasks (baselines), as well as the contrast between RC and the Q-critic. We will provide detailed operational steps in the appendix.
>
> [1] Guozheng Ma, Lu Li, Sen Zhang et al. Revisiting Plasticity in Visual Reinforcement Learning: Data, Modules and Training Stages. ICLR 2024

---

> > ### Author Rebuttal · Reviewer_VnXb · 2026-04-03
> >
> > Thank you for the detailed response. However, considering that Q1, Q2, and Q3 still involve many important details that need to be properly incorporated into the main text or appendix, I am inclined to raise my rating to weak reject. At this stage, however, I do not think the paper has reached the level of acceptance.

---

> > > ### Author Response · Authors · 2026-04-04
> > >
> > > Thank you for confirming that your technical concerns are "fully resolved" and for raising your rating. We completely agree with your remaining hesitation: a strong rebuttal is only a promise, and these profound insights must be integrated into the manuscript to stand alone.
> > >
> > > Fortunately, ICML policies allow for **an additional two pages** in the camera-ready version, providing ample space to incorporate these details comprehensively. To assure you of our commitment, here are the specific and comprehensive **draft texts** we have prepared for the final manuscript:
> > >
> > > [Draft Text for **Q1 TD limitations & Q2 Literature**]
> > >
> > > Insert before 'To avoid representation information overlap...' in the introduction:
> > >
> > > "Under an ideal MDP, TD learning gradually converges to the true expected return. However, this assumption often fails in VRL. As recent studies indicate, VRL faces significant difficulties in state representation learning due to the redundancy of high-dimensional pixel inputs (De Oliveira et al., Sliding Puzzles Gym:..., ICML 2025; Zhao et al., Reasoning as Representation: ..., ICLR 2026), which is further compounded by severe error accumulation caused by TD bootstrapping (Nauman et al., Overestimation, Overfitting..., ICML 2024; Wang et al., Making Offline RL Online:..., NeurIPS 2024). To explicitly illustrate this, we plot the n-step TD error (\~1) alongside the true long-term cumulative value prediction error (\~40) for a given state during training. Strikingly, while the n-step TD error quickly minimizes, a massive gap remains in the true long-term prediction error. This reveals a fundamental issue: TD heavily biases the visual encoder towards local features associated only with proximal immediate rewards, failing its true theoretical goal. Figure 3 provides intuitive visual evidence for this discrepancy: the TD-driven Q-Critic focuses merely on ground contact points, entirely missing the robot's joints that actually determine the long-term trajectory. Our Return-Critic directly mitigates this 'goal discrepancy' by providing a global, true supervision signal for the full episode, explicitly constraining the visual encoder to capture long-term return-relevant representations."
> > >
> > > Inserted before the final sentence of the Efficient Visual Reinforcement Learning:
> > >
> > > "Furthermore, recent advancements have explicitly utilized return signals to condition sequence modeling (Wang et al., Return Augmented..., NeurIPS 2025) or decouple reward-related features for offline reinforcement learning (Yang et al., Q-Supervised Contrastive Representation:..., ICML 2025). However, these methods often rely on static offline datasets or localized horizons, leaving the fundamental goal discrepancy in continuous online TD learning unaddressed."
> > >
> > > [Draft Text for **Q3 Baselines**]
> > >
> > > Inserted at the end of the Experiment Baselines:
> > >
> > > "Following recent findings that data augmentation primarily alleviates network plasticity loss, which is complementary to RC’s goal of improving representations, our primary evaluations integrate RC with the augmented baseline. Furthermore, to verify RC's fundamental effectiveness independent of data augmentation, we also provide evaluations based on the foundational SAC baseline."
> > >
> > > **Table A: Performance comparison based on the SAC baseline.**
> > >
> > > | Environment (1M Steps) | SAC | TACO w/ SAC | RC w/ SAC |
> > > | :--- | :---: | :---: | :---: |
> > > | Cheetah Run |  549±41 | 682±52 | **755±32** |
> > > | Walker Run | 312±26 | 467±34 | **608±22** |
> > > | Quadruped Run | 156±11 | 424±41 | **546±28** |
> > > | **Average Score** | 339.0 | 524.3 | **636.3** |
> > >
> > > "As shown in Table A, even without data augmentation, RC achieves a substantial performance leap, proving that our return-prediction objective inherently improves representation quality."
> > >
> > > [Draft Text for **Q4 Saliency Maps**]
> > >
> > > Insert before Figure 1:
> > >
> > > "To visually interpret the learned representations, we utilize Guided Backpropagation to project the encoder's last-layer gradients back to the input observation, generating high-resolution saliency heatmaps that highlight the features most causally related to the objective."
> > >
> > > **Conclusion:**
> > >
> > > We are deeply grateful for your rigorous review, which has genuinely elevated the quality and depth of this paper. The core contribution of our work lies in fundamentally identifying the "goal discrepancy" problem within existing TD and auxiliary task methods in VRL, and proposing a novel framework capable of explicitly focusing on return-relevant features. We have now thoroughly validated this solution both theoretically and empirically.
> > >
> > > Since you have kindly confirmed that all your technical concerns are fully resolved, we sincerely hope these concrete draft additions assure you that the final manuscript will be rigorous and self-contained. We would be deeply grateful if our commitment to these revisions might encourage you to view our paper favorably and consider supporting its acceptance. Thank you once again for your invaluable guidance!

---

### Official Review · Reviewer_r2d3 · 2026-03-13

**Soundness:** 3
**Presentation:** 3
**Significance:** 3
**Originality:** 3
**Overall Recommendation:** 3
**Confidence:** 4

**Summary:**

This paper proposes Return-Critic (RC) for visual RL. The main idea is to train a shared visual encoder with an episode-level return prediction objective, so that learned representations are more aligned with long-term return. The method also uses attention to learn a key function for prioritized replay. Results on DMControl and V-D4RL show consistent improvements over strong baselines.

**Compliance With Llm Reviewing Policy:**

Affirmed.

**Final Justification:**

This paper proposes Return-Critic (RC), a visual RL framework that aligns representation learning with long-term returns via an episode-level return prediction objective, and introduces an attention-based key function for prioritized replay, achieving consistent gains across DMControl and V-D4RL.

While the method is well-motivated and empirically strong, the contribution of each component and the theoretical grounding of the prioritized replay mechanism remain insufficiently clarified, so I maintain my current score.

**Key Questions For Authors:**

1. Since RC roughly doubles wall-clock time over DrQv2, can the authors report matched wall-clock or matched update-budget comparisons?
2. To what extent does the theory justify the full method? In particular, does it also support the key-function-based prioritized replay mechanism, or is it mainly intended to explain the representation-learning benefit of the return-prediction objective?

**Limitations:**

yes

**Strengths And Weaknesses:**

Strengths:
The paper is well motivated, and the core idea is simple and intuitive. I like the goal-discrepancy perspective, and the method is modular enough to be added to existing off-policy visual RL algorithms. The experimental results are also strong in both online and offline settings, which makes the paper practically interesting.

Weakness:
First, the source of the performance gain is still not fully disentangled, especially the relative contributions of the return prediction objective for representation learning and the key-function-based prioritized replay mechanism.
Secondly,the theory mainly explains the return-prediction objective, but it is less clear whether it also justifies the key-function-based prioritized replay part of the method.

---

> ### Author Rebuttal · Authors · 2026-03-28
>
> Thank you for recognizing the motivation and experimental results of RC. We have addressed your questions and the weaknesses you pointed out one by one below.
>
> Response to **Question 1**.
>
> As shown in Table 6 in the Appendix, the training wall-clock time of RC is indeed higher than that of the base algorithm DrQv2. However, this additional time investment yields a massive 68% improvement in average task scores (as shown in Figure 4, where the score improvement exceeds 100% in certain challenging tasks). Compared to similar SOTA auxiliary task methods, RC's wall-clock time is comparable to TACO (about 12 hours) , yet RC achieves a significant 20.2% lead in average task scores. Furthermore, we analyzed the computational resource footprint of these methods. Evaluated on the same hardware environment, RC consumes only 2.6GB of VRAM. While this is slightly higher than the base DrQv2 (1.7GB), it is more memory-efficient than TACO (5.1GB) and significantly lighter than other baseline methods such as MIND (18.0GB), PlayVirtual (16.2GB) and MLR (over 24GB). Therefore, under the same computational time and resource budget, RC holds an absolute advantage in both sample efficiency and final performance. More details will be provided in the appendix.
>
> Response to **Question 2**.
>
> Your insight is accurate. Our theoretical analysis in Section 3.3 primarily proves that the "return prediction" objective effectively bridges the Goal Discrepancy, while the "prioritized sampling mechanism" currently relies on empirical intuition rather than a formal mathematical proof. This design is fundamentally based on the Transformer's inherent Credit Assignment capability, which automatically focuses on discriminative inputs contributing to both high and low returns. As demonstrated in Figure 6 and validated by recent work (AttnRL [1]), attention weights naturally focus on positions with decisive impacts on the final outcome. Therefore, extracting the normalized attention score $\alpha$ as a pseudo-label to measure sample importance is theoretically intuitive and empirically effective for optimizing the sampling process.
>
> To ensure the full method functions reliably in practice, we implemented rigorous mechanisms for training stability. Since $\alpha$ is obtained by normalizing attention weights, it naturally stabilizes as the auxiliary task converges(empirically within a short span of 10K steps). Furthermore, when calculating the loss for Eq. (7), we apply a detach (stop-gradient) operation to the target score $\alpha$ to prevent disrupting the encoder, alongside strict Gradient Clipping during backpropagation to eliminate risks of numerical overflow. The rapidly stabilizing $\alpha$ combined with these engineering designs collectively guarantees the stable optimization of our key function. A more in-depth theoretical derivation specifically addressing the prioritized sampling mechanism will be added in Appendix.
>
> Response to **Weakness 1**.
>
> We fully agree on the importance of disentangling these components. Our analysis (in Table 4) demonstrates that the return prediction objective (representation learning) acts as the primary driver of the performance leap, while the key-score prioritized replay provides a crucial secondary boost. Quantitatively, the disentangled ablation in Table 4 shows that the "w/o $\mathcal{L}{key}$" variant (reverting to random sampling) still achieves massive gains relying solely on representation learning. Guided Backpropagation heatmaps (Figures 1 and 3) reveal that this objective effectively constrains the encoder to focus on visual features directly causally related to the total return, globally aligning with RL's ultimate goal. Furthermore, the specific performance gap between Full RC and "w/o $\mathcal{L}_{key}$" isolates the contribution of prioritized replay: by utilizing attention scores to actively oversample critical transitions, it prevents the encoder from overfitting to frequent but uninformative states. Ultimately, these mechanisms are highly synergistic: return prediction provides robust base representations and importance pseudo-labels, which prioritized replay leverages to optimize training data efficiency, jointly delivering the overall performance leap.
>
> Response to **Weakness 2**.
>
> As stated in our response to Question 2, we acknowledge that our current theoretical derivations (Section 3.3) primarily serve the representation learning component. The prioritized sampling mechanism currently serves as an empirical extension, functioning based on the Transformer's natural Credit Assignment capability. To ensure academic rigor, we will explicitly delineate the "applicable boundaries of the theoretical guarantees" from the "empirical and intuitive basis for prioritized sampling" in the revised version.
>
> [1] Runze Liu, Jiakang Wang, Yuling Shi et al. Attention as a Compass: Efficient Exploration for Process-Supervised RL in Reasoning Models. ICLR 2026

---

> > ### Author Rebuttal · Reviewer_r2d3 · 2026-04-02
> >
> > Thanks for the authors' explanation. My questions are mostly addressed. However, your response does not provide matched wall-clock or matched update-budget comparisons. Can you clarify whether RC still outperforms DrQv2 under equal training time or equal number of updates? Otherwise, how can we rule out that the reported gains are primarily due to increased computation?

---

> > > ### Author Response · Authors · 2026-04-04
> > >
> > > Thank you for your prompt acknowledgement and for giving us the opportunity to further address your concerns. We sincerely apologize for not accurately grasping the core of your question in our previous response.
> > >
> > > To directly answer your question: RC significantly outperforms DrQv2 and TACO under strictly equal update-budgets and strictly equal wall-clock time. We can confidently rule out that the reported gains are due to more updates or time. After careful internal discussion, we would like to specifically clarify the experimental settings in our paper to address the update-budget comparison, and supplement experimental results to intuitively demonstrate the performance of different methods under identical wall-clock time limits:
> > >
> > > 1. Clarification on **Matched Update-Budget** Benchmark:
> > >
> > > Our standard 1M-step experiments presented in the paper are already strictly matched in terms of update-budget. In our experimental setup, the gradient update frequency is directly tied to the environment steps. Specifically, DrQv2, TACO, and our RC method all utilize an identical update frequency of 2 (i.e., one gradient update every two environment steps). Therefore, **under the 1M-step setting, DrQv2, TACO, and RC method execute exactly 500K updates**. As demonstrated by the existing results in our paper, under this strictly identical budget of 500K updates (1M steps), RC achieves a massive 68% improvement over DrQv2 and explicitly outperforms the state-of-the-art TACO by 20%.
> > >
> > > 2. Supplementary Experiments on **Matched Wall-Clock Time** Benchmark:
> > >
> > > Evaluating under strict wall-clock time limits explicitly isolates the true algorithmic efficiency. We supplemented experiments restricting DrQv2, TACO, and RC to strict 6-hour and 12-hour wall-clock time budgets across 9 challenging environments.
> > >
> > > Note: Because DrQv2 is a lightweight baseline without auxiliary tasks, it completes roughly 1M steps in 6 hours and 2M steps in 12 hours. RC and TACO complete roughly 500K steps in 6 hours and 1M steps in 12 hours.
> > >
> > > **Table A: Performance under equal 6-Hour Wall-Clock Time limit**
> > > *(Note: In 6 hours, DrQv2 completes ~1M steps, while RC & TACO complete ~0.5M steps)*
> > >
> > > | Environment (6 Hours) | DrQv2 (~1M steps) | TACO (~0.5M steps) | DrQv2 w/ RC (~0.5M steps) |
> > > | :--- | :---: | :---: | :---: |
> > > | Quadruped Run | 407±21 | 392±27 | **523±36** |
> > > | Hopper Hop | 189±35 | 154±41 | **270±48** |
> > > | Walker Run | 517±43 | 427±49 | **677±24** |
> > > | Quadruped Walk | **680±52** | 452±96 | 648±63 |
> > > | Cheetah Run | 691±42 | 644±34 | **817±22** |
> > > | Finger Turn Hard | 220±21 | **411±57** | 354±69 |
> > > | Acrobot Swingup | 128±8  | 102±15 | **238±46** |
> > > | Reacher Hard | 572±51 | **623±62** | 483±55 |
> > > | Reach Duplo | 206±32 | 202±11| **212±8** |
> > > | **Average Score** | 401.1  | 378.6 | **469.1** |
> > >
> > >
> > > **Table B: Performance under equal 12-Hour Wall-Clock Time limit**
> > > *(Note: In 12 hours, DrQv2 completes ~2M steps, while RC & TACO complete ~1M steps)*
> > >
> > > | Environment (12 Hours) | DrQv2 (~2M steps) | TACO (~1M steps) | DrQv2 w/ RC (~1M steps) |
> > > | :--- | :---: | :---: | :---: |
> > > | Quadruped Run | 494±73 |  541±38  | **714±52** |
> > > | Hopper Hop | 247±25 |  261±52  | **340±43** |
> > > | Walker Run | 629±31 | 637±11 | **736±14** |
> > > | Quadruped Walk | 867±12 | 793±8  | **875±15** |
> > > | Cheetah Run | 822±10 | 821±48  | **897±12** |
> > > | Finger Turn Hard | 653±64 | 632±75 | **913±55** |
> > > | Acrobot Swingup | 323±21 | 241±21 | **450±70** |
> > > | Reacher Hard | 898±14 |  883±63 | **913±45** |
> > > | Reach Duplo | 220±7 | **234±21** | 228±7 |
> > > | **Average Score** | 572.6 | 560.3 | **673.9** |
> > >
> > > The results demonstrate RC's robust superiority under both time constraints: **Under the 6-Hour Limit (Table A)**: Although RC (\~0.5M steps) has not yet reached its convergence potential within this shorter timeframe, it still achieves the best performance in most tasks. Strikingly, despite executing only half the environment interactions and updates of DrQv2 (\~1M steps), RC achieves a 17% higher average score (469.1 vs. 401.1). Meanwhile, the prior advanced method, TACO, loses its baseline advantage. **Under the 12-Hour Limit (Table B)**: With a longer time limit, DrQv2 (\~2M steps) fully plateaus at its asymptotic performance limit, eventually surpassing TACO (572.6 vs. 560.3). In stark contrast, RC (\~1M steps) still maintains an 18% performance advantage (673.9 vs. 572.6) over the fully converged DrQv2 (\~2M steps). This clearly shows that DrQv2 cannot bridge the performance gap with RC simply by training for a longer time.
> > >
> > > We sincerely appreciate this valuable follow-up question. RC maintains a dominant advantage under both matched update-budgets and wall-clock times, definitively proving our gains are not from extra computation. We hope these comprehensive new evaluations fully resolve your concerns, and we would be honored if you would consider raising your score in light of these results.

---

### Decision · Program_Chairs · 2026-04-30

**Decision:**

Accept (regular)

**Comment:**

Metareview Summary:
This paper received ratings of 4, 3, 3, and 3. After rebuttal, Reviewer 4W1q's concerns were largely addressed, while Reviewers VnXb, r2d3, and NXir maintained reservations about text clarification, disentangling contributions/theoretical grounding, and offline dataset sensitivity to data coverage.

AC Assessment:
The AC respectfully disagrees with key reviewer concerns:
1. Disentangling contributions/Theoretical grounding (Reviewer r2d3): Table 4 clearly isolates the prioritized replay contribution via the "w/o L_key" ablation, demonstrating that representation learning alone drives substantial gains. The AC considers the theoretical justification for the goal-discrepancy framework adequate and does not require additional formal proof for the prioritized sampling mechanism, which relies on sound intuitive foundations grounded in Transformer credit assignment.
2. Return capture in standard RL (Reviewer VnXb):  Predicting returns from full episode trajectories provides explicit global supervision, fundamentally different from single-frame value estimation. This approach captures return-relevant visual representations by bypassing TD bootstrapping errors, enabling the encoder to focus on features directly causal to overall episode returns.
3. Offline dataset coverage (Reviewer NXir): Suboptimal expert dataset performance reflects the base algorithm's limitation, not RC's framework. RC is designed primarily for online RL, where exploration naturally generates diverse trajectories with sufficient return variance.

Recommendation:
The core contributions are technically sound and well-supported by extensive rebuttal experiments (matched wall-clock/update-budget comparisons, state-based MuJoCo results, comprehensive ablations). Author commitments to camera-ready improvements align with ICML's revision policy.

Final Recommendation: Accept (Weak Accept)